# Coded Computing for Resilient Distributed Computing: A Learning-Theoretic Framework

**Parsa Moradi**
University of Minnesota
moradi@umn.edu

**Behrooz Tahmasebi**
MIT CSAIL
bzt@mit.edu

**Mohammad Ali Maddah-Ali**
University of Minnesota
maddah@umn.edu

## Abstract

Coded computing has emerged as a promising framework for tackling significant challenges in large-scale distributed computing, including the presence of slow, faulty, or compromised servers. In this approach, each worker node processes a combination of the data, rather than the raw data itself. The final result then is decoded from the collective outputs of the worker nodes. However, there is a significant gap between current coded computing approaches and the broader landscape of general distributed computing, particularly when it comes to machine learning workloads. To bridge this gap, we propose a novel foundation for coded computing, integrating the principles of learning theory, and developing a framework that seamlessly adapts with machine learning applications. In this framework, the objective is to find the encoder and decoder functions that minimize the loss function, defined as the mean squared error between the estimated and true values. Facilitating the search for the optimum decoding and functions, we show that the loss function can be upper-bounded by the summation of two terms: the generalization error of the decoding function and the training error of the encoding function. Focusing on the second-order Sobolev space, we then derive the optimal encoder and decoder. We show that in the proposed solution, the mean squared error of the estimation decays with the rate of $\mathcal{O}(S^3 N^{-3})$ and $\mathcal{O}(S^{8/5} N^{-3/5})$ in noiseless and noisy computation settings, respectively, where $N$ is the number of worker nodes with at most $S$ slow servers (stragglers). Finally, we evaluate the proposed scheme on inference tasks for various machine learning models and demonstrate that the proposed framework outperforms the state-of-the-art in terms of accuracy and rate of convergence.

## 1 Introduction

The theory of *coded computing* has been developed to improve the reliability and security of large-scale machine learning platforms, effectively tackling two major challenges: (1) the detrimental impact of *slow workers (stragglers)* on overall computation efficiency, and (2) the threat of *faulty or malicious workers* that can compromise data accuracy and integrity. These challenges have been well-documented in the literature, including the seminal work [1] from Google. For instance, [2] reported that in a sample set of 3000 matrix multiplication jobs on AWS Lambda, while the median job time was 40 seconds, approximately 5% of worker nodes took 100 seconds to respond, and two nodes took as long as 375 seconds. Furthermore, coded computing has also been instrumental in addressing *privacy concerns*, a crucial aspect of distributed computing systems [3–12].

The concept of coded computing has been motivated by the success of coding in communication over unreliable channels, where instead of transmitting raw data, the transmitter sends a (linear) combination of the data, known as coded data. This redundancy in the coded data enables the

38th Conference on Neural Information Processing Systems (NeurIPS 2024).

receiver to recover the raw data even in the presence of errors or missing values. Similarly, coded computing includes three layers [3, 11, 13] (see Figure 1(a)):

(1) *The Encoding Layer* in which a master node sends a (linear) combination of data, as coded data, to each worker node.

(2) *The Computing Layer*, in which the worker nodes apply a predefined computation to their assigned coded data and send the results back to the master node.

(3) *The Decoding Layer*, in which the master node recovers the final results from the computation results over coded data. In this layer, the decoder leverages the coded redundancy in the computation to recover the missing results of the stragglers and detect and correct the adversarial outputs.

The existing coded computing has largely built upon algebraic coding theory, drawing inspiration from the renowned Reed-Solomon code construction in communication [14], with proven straggler and Byzantine resiliency [15]. However, the coding in communication is designed for the exact recovery of the messages, built on a foundation that is inconsistent with the computational requirements of machine learning. Developing a code that preserves its specific construction while composing with computation is extremely challenging, leading to significant restrictions. Firstly, current methods are mainly restricted to specific computation functions, such as polynomials and matrix multiplication [3, 11, 13, 16, 17]. Secondly, rooted in algebraic error correction codes, existing approaches are tailored for finite field computations, leading to numerical instability when dealing with real-valued data [18, 19]. Furthermore, these methods are unsuitable for approximate, fixed-point, or floating-point computing, where exact computation is neither possible nor necessary, such as in machine learning inference or training tasks. Finally, these schemes typically have a recovery threshold, which is the minimum number of samples required to recover results from coded outputs of worker nodes [3, 13]. If the number of workers falls below this threshold, the recovery process fails entirely.

Several works have attempted to mitigate the aforementioned issues and transform the coded computing scheme into a more robust and adaptable one, applicable to a wide range of computation functions. These efforts include approximating non-polynomial functions with polynomial ones [5, 20], refining the coding mechanism to enhance stability [21–25], and leveraging approximation computing techniques to reduce the recovery threshold and increase recovery flexibility [26–29]. However, these attempts fail to bridge the existing gap between coded computing and general distributed computing systems. The root cause of these issues lies in the fact that they are grounded in coding theory, based on a foundation that is not compatible with the requirements of large-scale machine learning. Therefore, this paper aims to address the following objective:

> **Objective:** The main objective of this paper is to develop a new foundation for coded computing, not solely based on *coding theory*, but also grounded in *learning theory*, that seamlessly integrates with machine learning applications, offering a more natural and effective solution for *general computing*.

In this paper, we establish a learning-theoretic foundation for coded computing, applicable to general computations. We adopt an end-to-end system perspective, that integrates an end-to-end loss function, to find the optimum encoding and decoding functions, focusing on straggler resiliency. We show that the loss function is upper-bounded by the sum of two terms: one characterizing the *generalization error* of the decoder function and the other capturing the *training error* of the encoder function. Regularizing the decoder layer, we derive the optimal decoder in the Reproducing Kernel Hilbert space (RKHS) of second-order Sobolev functions. This provides an explicit solution for the optimum decoder function and allows us to characterize the resulting loss of the decoding layer. The decoder loss appears as a regularizing term in optimizing the encoding function and represents the norm in another RKHS. Thus, the optimum solution for the encoding function can be derived, too. We address two noise-free and noisy computation settings, for which we derive the *optimal encoder and decoder* and corresponding convergence rate. We prove that the proposed framework exhibits a faster convergence rate compared to the state-of-the-art and the numerical evaluations support the theoretical derivations (see Figure 1(b)).

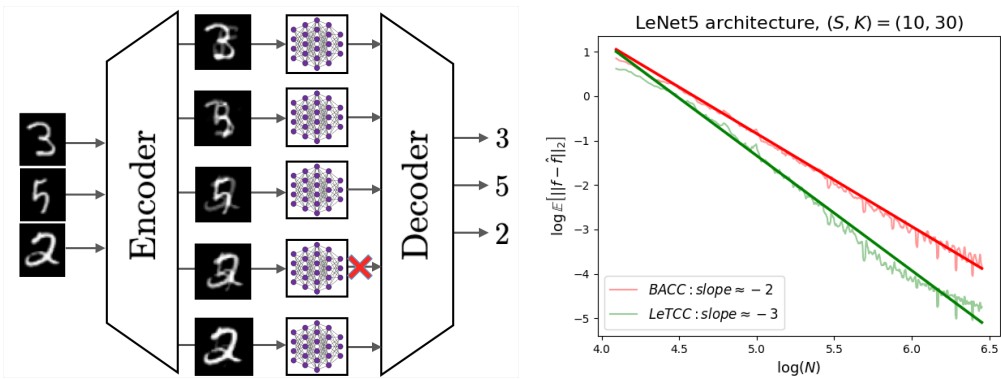

Figure 1(a): Coded Computing: Each worker node processes a combination of data (coded data). The decoder recovers the final results, even in the presence of missing outputs from some worker nodes.

Figure 1(b): The log-log plot of the expected error versus the number of workers ($N$) for the proposed framework (LeTCC) and the state-of-the-art BACC [29]. LeTCC framework not only achieves a lower estimation error but also has a faster convergence rate.

**Contributions:** The main contributions of this paper are:

- We develop a new foundation for coded computing by integrating it with learning theory, rather than relying solely on coding theory. We define the loss function as the mean square error of the computation estimation, averaged over all possible sets of at most $S$ stragglers (Section 3.1). To be able to find the best encoding and decoding functions, we bound the loss function with the summation of two terms, one characterizing the generalization error of the decoder function and the other capturing the training error of the encoder function (Section 3).

- Assuming that the encoder and decoder functions reside in the Hilbert space of second-order Sobolev functions, we use the theory of RKHSs to find the optimum encoding and decoding functions and characterize the convergence rate for the expected loss in both noise-free and noisy computation regimes (Section 4).

- We have extensively evaluated the proposed scheme across different data points and computing functions including state-of-the-art deep neural networks and demonstrated that our proposed framework considerably outperforms the state-of-the-art in terms of recovery accuracy (Section 5).

## 2 Preliminaries and Problem Definition

### 2.1 Notations

Throughout this paper, uppercase and lowercase bold letters denote matrices and vectors, respectively. Coded vectors and matrices are indicated by a $\sim$ sign, as in $\tilde{\mathbf{x}}, \tilde{\mathbf{A}}$. The set $\{1, 2, \ldots, n\}$ is denoted as $[n]$ and symbol $|S|$ denotes the cardinality of the set $S$. Finally, we represent first, second and $k$-th order derivative of function $f$ as $f'$, $f''$, and $f^{(k)}$, respectively.

### 2.2 Problem Setting

Consider a master node and a set of $N$ workers. The master node is tasked with computing $\{\mathbf{f}(x_k)\}_{k=1}^{K}$ using a cluster of $N$ worker nodes, given a set of $K$ data points $\{\mathbf{x}_k\}_{k=1}^{K}, \mathbf{x}_k \in \mathbb{R}^d$. Here, $\mathbf{f} : \mathbb{R}^d \to \mathbb{R}^m$ represents an arbitrary function, which could be a simple one-dimensional function or a complex deep neural network, and $K, d, m$ are integers. A naive approach would be to assign the computation of $\mathbf{f}(\mathbf{x}_k)$ to one worker node for $k \in [K]$. However, some worker nodes may act as stragglers, failing to complete their tasks within the required deadline. To mitigate this issue, the master node employs coding and sends $N$ coded data points to each worker node using an encoder function. Each coded data point is a combination of raw data points. Subsequently, each

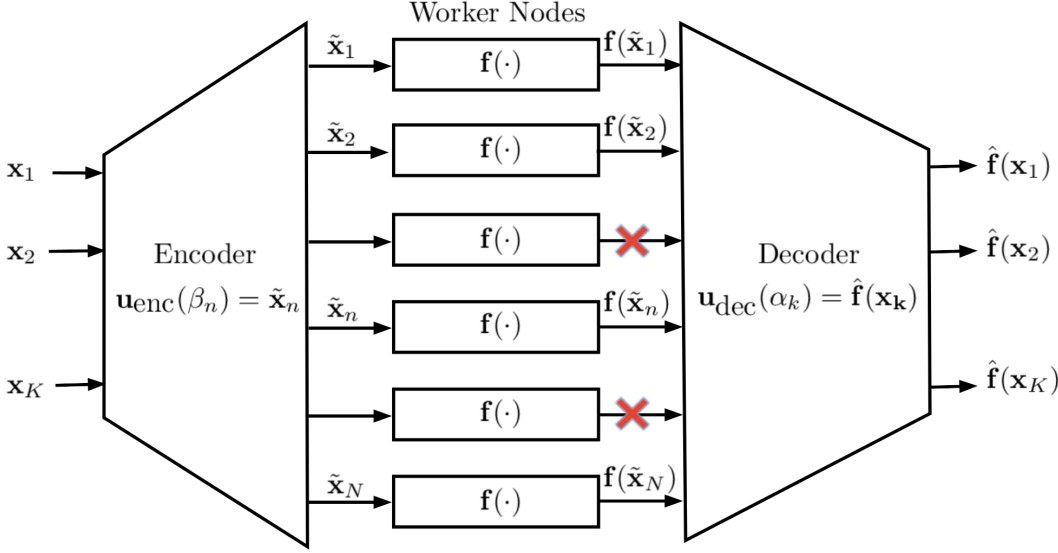

Figure 2: LeTCC framework.

worker applies the function $\mathbf{f}(\cdot)$ to the received coded data and sends the result, coded results, back to the master node. The master node's goal is to approximately recover $\hat{\mathbf{f}}(\mathbf{x}_k) \approx \mathbf{f}(\mathbf{x}_k)$ using a decoder function, even if some worker nodes appear to be stragglers. The redundancy in the coded data and corresponding coded results enables the master node to recover the desirable results, $\{\mathbf{f}(\mathbf{x}_k)\}_{k=1}^{K}$.

## 3 Proposed Framework: LeTCC

Here, we propose a novel straggler-resistant Learning-Theoretic Coded Computing (LeTCC) framework for general distributed computing. As depicted in Figure 2, our framework comprises two encoding and decoding layers, with a computing layer sandwiched between them. The framework operates according to the following steps:

(1) **Encoding Layer:** The master node fits an encoder regression function $\mathbf{u}_{\mathrm{enc}} : \mathbb{R} \to \mathbb{R}^d$ at points $\{(\alpha_k, \mathbf{x}_k)\}_{k=1}^{K}$ for fixed, distinct, and ordered values $\alpha_1 < \alpha_2 < \cdots < \alpha_K \in \mathbb{R}$. Then, it computes the encoder function $\mathbf{u}_{\mathrm{enc}}(\cdot)$ on another set of fixed, distinct, and ordered values $\{\beta_n\}_{n=1}^{N}$ where $\beta_1 < \beta_2 < \cdots < \beta_N \in \mathbb{R}$, with $k \in [K]$ and $n \in [N]$. Subsequently, the master node sends the coded data points $\tilde{\mathbf{x}}_n = \mathbf{u}_{\mathrm{enc}}(\beta_n) \in \mathbb{R}^d$ to worker $n$ for $n \in [N]$. Note that each coded data point $\tilde{\mathbf{x}}_n$ is a combination of all initial points $\{\mathbf{x}_k\}_{k=1}^{K}$.

(2) **Computing Layer:** Each worker node $n \in [N]$ computes $\mathbf{f}(\tilde{\mathbf{x}}_n) = \mathbf{f}(\mathbf{u}_{\mathrm{enc}}(\beta_n))$ on its assigned input and sends the result back to the master node.

(3) **Decoding Layer:** The master node receives the results $\mathbf{f}(\tilde{\mathbf{x}}_v)_{v \in \mathcal{F}}$ from the non-straggler worker nodes in the set $\mathcal{F}$. Next, it fits a decoder regression function $\mathbf{u}_{\mathrm{dec}} : \mathbb{R} \to \mathbb{R}^m$ at points $(\beta_v, \mathbf{f}(\tilde{\mathbf{x}}_v))_{v \in \mathcal{F}} = (\beta_v, \mathbf{f}(\mathbf{u}_{\mathrm{enc}}(\beta_v)))_{v \in \mathcal{F}}$. Finally, using the function $\mathbf{u}_{\mathrm{dec}}(\cdot)$, the master node computes $\hat{\mathbf{f}}(\mathbf{x}_k) := \mathbf{u}_{\mathrm{dec}}(\alpha_k)$ as an approximation of $\mathbf{f}(\mathbf{x}_k)$ for $k \in [K]$. Recall that $\mathbf{u}_{\mathrm{dec}}(\alpha_k) \approx \mathbf{f}(\mathbf{u}_{\mathrm{enc}}(\alpha_k)) \approx \mathbf{f}(\mathbf{x}_k)$.

As mentioned above, the master node selects and fixes the regression points, $\{\alpha_k\}_{k=1}^{K}$ and $\{\beta_n\}_{n=1}^{N}$, which remain constant throughout the entire process. The encoder and decoder functions are the only components subject to optimization.

Note that the computational efficiency of the encoding and decoding layers is crucial. This includes the fitting process of the encoder and decoder regression functions, as well as the computation of these regression functions at points $\{\beta_v\}_{v \in \mathcal{F}}$ and $\{\alpha_k\}_{k=1}^{K}$. If the master node's computation time is not substantially decreased compared to computing $\{\mathbf{f}(\mathbf{x}_k)\}_{k=1}^{K}$ by itself, then adopting this framework would not provide any benefits for the master node.

## 3.1 Objective

We view the whole scheme as a unified predictive framework that provides an approximate estimation of the values $\{\mathbf{f}(\mathbf{x}_k)\}_{k=1}^K$. We denote the estimator function of the LeTCC scheme as $\hat{\mathbf{f}}_{\boldsymbol{\alpha},\boldsymbol{\beta}}[\mathbf{u}_{\text{enc}}, \mathbf{u}_{\text{dec}}, \mathcal{F}](\cdot)$, where $\boldsymbol{\alpha} := [\alpha_1, \ldots, \alpha_K]^T$, $\boldsymbol{\beta} := [\beta_1, \ldots, \beta_N]^T$, and $\mathcal{F} := \{i_1, \ldots, i_{|\mathcal{F}|}\}$ represents the set of non-straggler worker nodes.

Let us define a random variable $F_{S,N}$ distributed over the set of all subsets of $N$ workers with maximum $S$ stragglers, $\{\mathcal{F} : \mathcal{F} \subseteq [N], |\mathcal{F}| \geqslant N - S\}$. Also, suppose each worker node $n \in [N]$ computes the function $\mathbf{f}_n(x) = \mathbf{f}(x) + \boldsymbol{\epsilon}_n$, where $\boldsymbol{\epsilon}_n$, $n \in [N]$ are independent zero-mean noise vectors with covariance $\sigma^2 \mathbf{I}$.

This enables us to define the following loss function, which evaluates the framework's performance:

$$\mathcal{R}(\hat{\mathbf{f}}) := \mathop{\mathbb{E}}_{\boldsymbol{\epsilon}, \mathcal{F} \sim F_{S,N}} \left[ \frac{1}{K} \sum_{k=1}^K \left\| \hat{\mathbf{f}}(\mathbf{x}_k) - \mathbf{f}(\mathbf{x}_k) \right\|_2^2 \right] = \mathop{\mathbb{E}}_{\boldsymbol{\epsilon}, \mathcal{F} \sim F_{S,N}} \left[ \frac{1}{K} \sum_{k=1}^K \| \mathbf{u}_{\text{dec}}(\alpha_k) - \mathbf{f}(\mathbf{x}_k) \|_2^2 \right], \quad (1)$$

where $\hat{\mathbf{f}}(\mathbf{x}) := \hat{\mathbf{f}}_{\boldsymbol{\alpha},\boldsymbol{\beta}}[\mathbf{u}_{\text{enc}}, \mathbf{u}_{\text{dec}}, \mathcal{F}](\mathbf{x})$ to simplify the notation, $\|\cdot\|_2$ represents the $\ell_2$-norm, and $\boldsymbol{\epsilon} = [\boldsymbol{\epsilon}_1, \ldots, \boldsymbol{\epsilon}_N]^T$. Our objective is to find $\mathbf{u}_{\text{enc}}(.)$ and $\mathbf{u}_{\text{dec}}(.)$ that minimize the objective function (1), which is very challenging, given that $\hat{\mathbf{f}}(.)$ is a composition of $\mathbf{u}_{\text{enc}}(.)$ and $\mathbf{u}_{\text{dec}}(.)$ and the computation in the middle. Here, we take an important step to decompose these two, to gain a deeper understanding of interactions. Adding and subtracting $\mathbf{f}(\mathbf{u}_{\text{enc}}(\alpha_k))$ and utilizing inequality of arithmetic and geometric means (AM-GM), one can obtain an upper bound for (1):

$$\mathcal{R}(\hat{\mathbf{f}}) = \mathop{\mathbb{E}}_{\boldsymbol{\epsilon}, \mathcal{F} \sim F_{S,N}} \left[ \frac{1}{K} \sum_{k=1}^K \| (\mathbf{u}_{\text{dec}}(\alpha_k) - \mathbf{f}(\mathbf{u}_{\text{enc}}(\alpha_k))) + (\mathbf{f}(\mathbf{u}_{\text{enc}}(\alpha_k)) - \mathbf{f}(\mathbf{x}_k)) \|_2^2 \right]$$

$$\leqslant \underbrace{\mathop{\mathbb{E}}_{\boldsymbol{\epsilon}, \mathcal{F} \sim F_{S,N}} \left[ \frac{2}{K} \sum_{k=1}^K \| \mathbf{u}_{\text{dec}}(\alpha_k) - \mathbf{f}(\mathbf{u}_{\text{enc}}(\alpha_k)) \|_2^2 \right]}_{\mathcal{L}_{\text{dec}}(\hat{\mathbf{f}})} + \underbrace{\frac{2}{K} \sum_{k=1}^K \| \mathbf{f}(\mathbf{u}_{\text{enc}}(\alpha_k)) - \mathbf{f}(\mathbf{x}_k) \|_2^2}_{\mathcal{L}_{\text{enc}}(\hat{\mathbf{f}})}. \quad (2)$$

The right-hand side of (2) comprises two terms, which uncover an interesting interplay between the encoder and decoder regression functions. Let us elaborate on what each term corresponds to.

- $\mathcal{L}_{\text{dec}}(\hat{\mathbf{f}})$ – **The expected generalization error of the decoder regression:** Recall that the master node fits a decoder regression function, $\mathbf{u}_{\text{dec}}(\cdot)$, at a set of points denoted as $\{(\beta_v, \mathbf{f}(\mathbf{u}_{\text{enc}}(\beta_v)))\}_{v \in \mathcal{F}}$. $\mathcal{L}_{\text{dec}}$ represents the $\ell_2$-norm of the decoder regression function's error on a distinct set of points $\{\alpha_k\}_{k=1}^K$, which are *different* from its training data $\{\beta_v\}_{v \in \mathcal{F}}$. Consequently, this term provides an unbiased estimate of the decoder's generalization error. Given that the decoder regression function develops to estimate $\mathbf{f}(\mathbf{u}_{\text{enc}}(\cdot))$, the generalization error of the decoder regression is inherently tied to the properties of $\mathbf{f}(\mathbf{u}_{\text{enc}}(\cdot))$. This, in turn, is influenced by characteristics of both the $\mathbf{f}(\cdot)$ and $\mathbf{u}_{\text{enc}}(\cdot)$ functions, making the $\mathcal{L}_{\text{dec}}(\hat{\mathbf{f}})$ a complex interplay of these two functions.

- $\mathcal{L}_{\text{enc}}(\hat{\mathbf{f}})$ – **A proxy to the training error of the encoder regression:** Remember that the encoder regression is fitted at points $\{(\alpha_k, \mathbf{x}_k)\}_{k=1}^K$. Consequently, the training error is calculated as $\frac{1}{K} \sum_{k=1}^K \| \mathbf{u}_{\text{enc}}(\alpha_k) - \mathbf{x}_k \|_2^2$. Therefore, $\mathcal{L}_{\text{enc}}$ represents the encoder training error magnified by the effect of computing function $\mathbf{f}(\cdot)$. Specifically, if $\mathbf{f}(\cdot)$ is $q$-Lipschitz, then $\mathcal{L}_{\text{enc}}(\hat{\mathbf{f}})$ can be upper bounded by:

$$\frac{2}{K} \sum_{k=1}^K \| \mathbf{f}(\mathbf{u}_{\text{enc}}(\alpha_k)) - \mathbf{f}(\mathbf{x}_k) \|_2^2 \leqslant \frac{2q^2}{K} \sum_{k=1}^K \| \mathbf{u}_{\text{enc}}(\alpha_k) - \mathbf{x}_k \|_2^2. \quad (3)$$

## 4   Main Results

In this section, we examine the proposed framework from a theoretical standpoint. We provide a comprehensive explanation of the design process for the decoder and encoder functions and subsequently analyze the convergence rate. For simplicity, we present the results for a one-dimensional

function $f : \mathbb{R} \to \mathbb{R}$. These results are generalizable to the case where $f : \mathbb{R} \to \mathbb{R}^m$, as discussed in Appendix F.

Suppose the regression points, $\{\alpha_k\}_{k=1}^K$, $\{\beta_n\}_{n=1}^N$, are confined to the interval $\Omega := (-1, 1)$ and $u_{\text{enc}}, u_{\text{dec}} \in \widetilde{\mathcal{H}}^2(\Omega; \mathbb{R})$, where $\widetilde{\mathcal{H}}^2(\Omega; \mathbb{R})$ is the reproducing kernel Hilbert space (RKHS) of second-order Sobolev functions on the interval $\Omega$ induced with the norm $\|f\|_{\widetilde{\mathcal{H}}^2(\Omega;\mathbb{R})}^2 := \int_\Omega (f''(t))^2 \, dt + f(-1)^2 + f'(-1)^2$ which is an equivalent norm on Sobolev space introduced by [30] (see (28) in Appendix A.1). The definition and properties of Sobolev spaces, along with their reproducing kernels and norms, are reviewed in Appendix A.1.

**Decoder Design:** Since $\mathcal{L}_{\text{dec}}(\hat{\mathbf{f}})$ in the decomposition (2) characterizes the generalization error of the decoder function, we propose a regularized objective function for the decoder:

$$u_{\text{dec}}^\star = \underset{u \in \widetilde{\mathcal{H}}^2(\Omega;\mathbb{R})}{\operatorname{argmin}} \frac{1}{|\mathcal{F}|} \sum_{v \in \mathcal{F}} (u(\beta_v) - f(u_{\text{enc}}(\beta_v)))^2 + \lambda_d \int_\Omega (u''(t))^2 \, dt. \tag{4}$$

The first term in (4) corresponds to the mean squared error, while the second term characterizes the smoothness of the decoder function on the interval $\Omega$. Equation (4) represents a Kernel Ridge Regression problem (KRR). It can be shown that the solution of (4) has the following form [31, 32]:

$$d_0 + d_1 t + \sum_{v=1}^{|\mathcal{F}|} c_v \phi_0(t, \beta_{i_v}), \tag{5}$$

where $d_0, d_1 \in \mathbb{R}$, $\phi_0(\cdot, \cdot)$ is the kernel function of $\mathcal{H}_0^2(\Omega; \mathbb{R})$ (see Definition 2 and (44) in Appendix A.1), and $\mathbf{c} = [c_1, \ldots, c_{|\mathcal{F}|}]^T \in \mathbb{R}^{|\mathcal{F}|}$. Substituting (5) into the main objective (4), the coefficient vectors $\mathbf{c}$ and $\mathbf{d} := [d_0, d_1]^T$ can be efficiently computed by optimizing a quadratic equation [33]. This solution is known as the *second-order smoothing spline* function. The theoretical properties of smoothing splines are reviewed in Appendix A.2.

Let us define the following variables, which represent the maximum and minimum distances between consecutive data points in the decoder layer, $\{\beta_n\}_{n=1}^N$:

$$\Delta_{\max} := \max_{n \in \{0\} \cup [N]} \{\beta_{n+1} - \beta_n\}, \quad \Delta_{\min} := \min_{n \in [N-1]} \{\beta_{n+1} - \beta_n\}, \tag{6}$$

with $\beta_0 := -1$ and $\beta_{N+1} := 1$. The following theorems provide crucial insights for designing the encoder function as well as deriving the convergence rates.

**Theorem 1** (Upper bound for noiseless computation, $\sigma_0 = 0$). *Consider the LeTCC framework with $N$ worker nodes and at most $S$ stragglers with $\lambda_d \leqslant N^{-4}$. Assume $\{\alpha_k\}_{k=1}^K$ are arbitrary and distinct points in $\Omega = (-1, 1)$ and there is constant $B$ such that $\frac{\Delta_{max}}{\Delta_{min}} \leqslant B$. If $f(\cdot)$ is a q-Lipschitz continuous function, then:*

$$\mathcal{R}(\hat{f}) \leqslant C_1 \left(\frac{S+1}{N}\right)^3 \cdot \|(f \circ u_{enc})''\|_{L^2(\Omega;\mathbb{R})}^2 + \frac{2q^2}{K} \sum_{k=1}^K (u_{enc}(\alpha_k) - x_k)^2, \tag{7}$$

*where $C_1$ is a constant.*

The proof of Theorem 1 and the detailed expression for $C_1$ can be found in Appendix B.1.

**Theorem 2** (Upper bound for noisy computation). *Consider the LeTCC framework with $N$ worker nodes and at most $S$ stragglers and $\frac{1}{(N-S)^4} \leqslant \lambda_d \leqslant \lambda_0$ for constant $\lambda_0 \in \mathbb{R}$. Assume each worker node computes $f_n(x) = f(x) + \epsilon_n$ with $\mathbb{E}[\epsilon_n] = 0$ and $\mathbb{E}[\epsilon_n^2] \leqslant \sigma_0^2$. Assume $\{\alpha_k\}_{k=1}^K$ are arbitrary and distinct points in $\Omega = (-1, 1)$ and suppose there is constant $B$ such that $\frac{\Delta_{max}}{\Delta_{min}} \leqslant B$. Assume $f(\cdot)$ is a q-Lipschitz continuous function. Then,*

$$\mathcal{R}(\hat{f}) \leqslant 4 \left(\frac{\sigma_0^2}{N-S}\right)^{\frac{3}{5}} \left(C_2 \cdot C(\lambda_0) \cdot p_4(S) \cdot \|(f \circ u_{enc})''\|_{L^2(\Omega;\mathbb{R})}^2\right)^{\frac{2}{5}} + \frac{2q^2}{K} \sum_{k=1}^K (u_{enc}(\alpha_k) - x_k)^2, \tag{8}$$

*if*

$$\frac{1}{(N-S)^{\frac{1}{5}}\lambda_0^{\frac{1}{4}}} \leqslant \left( \frac{C_2 \cdot C(\lambda_0) \cdot p_4(S) \cdot \left\| (f \circ u_{enc})^{(2)} \right\|_{L^2(\Omega;\mathbb{R})}^2}{\sigma_0^2} \right)^{\frac{1}{5}} \leqslant (N-S)^{\frac{4}{5}} \qquad (9)$$

*where $C_2$ is a constant, $C(\lambda_0) = \mathcal{O}(\lambda_0^{\frac{1}{2}})$ is an increasing function of $\lambda_0$, and $p_4(S)$ is a degree-4 polynomial in $S$ with positive constant coefficients.*

The proof and expressions for $C_2$ and $C(\lambda_0)$ are provided in Appendix B.2.

**Encoder Design:** The upper bounds established in Theorems 1, 2 hold for all $u_{\text{enc}} \in \widetilde{\mathcal{H}}^2(\Omega;\mathbb{R})$. However, they do not directly lead to a design for $u_{\text{enc}}(\cdot)$. To address this, we present the following theorem which bounds the $\|f \circ u_{\text{enc}}\|_{L^2(\Omega;\mathbb{R})}^2$, enabling us to construct $u_{\text{enc}}(\cdot)$ without compromising the convergence rate.

**Theorem 3.** *Consider a `LeTCC` scheme. Assume computing function $f(\cdot)$ is $q$-Lipschitz continuous and $\|f''\|_{L^\infty(\Omega;\mathbb{R})} \leqslant \nu$. Then:*

$$\mathcal{R}(\hat{f}) \leqslant \frac{2q^2}{K} \sum_{k=1}^{K} (u_{enc}(\alpha_k) - x_k)^2 + \lambda_e \cdot \psi\big( \|u_{enc}\|_{\widetilde{\mathcal{H}}^2(\Omega;\mathbb{R})}^2 \big), \qquad (10)$$

*for some monotonically increasing function $\psi : \mathbb{R}^+ \rightarrow \mathbb{R}^+$, where $\lambda_e$ is depending on $(N, S, \sigma_0, q, \nu)$.*

The proof can be found in Appendix B.3. By applying the representer theorem [34], we can deduce that the optimal encoder $u_{\text{enc}}(\cdot)$, which minimizes the right-hand side of (10) takes the form $u_{\text{enc}}(\cdot) = \sum_{k=1}^{K} z_k \phi(\alpha_k, \cdot)$, where $\mathbf{z} \in \mathbb{R}^K$, and $\phi$ is the kernel function of $\widetilde{\mathcal{H}}^2(\Omega;\mathbb{R})$, as discussed in Appendix A.2 and in (46). However, due to the non-linearity of $g(\cdot)$, calculating the values of the coefficients $\mathbf{z}$ is challenging. Nevertheless, we demonstrate that the coefficients can be efficiently derived under certain mild assumptions.

**Proposition 1.** *In the noiseless case, there exists $M \in \mathbb{R}$ that depends only on $\{\alpha_k\}_{k=1}^K$ and $\{x_k\}_{k=1}^K$, such that:*

*(i) If $\|u_{enc}\|_{\widetilde{\mathcal{H}}^2(\Omega;\mathbb{R})}^2 \leqslant M$, then:*

$$\mathcal{R}(\hat{f}) \leqslant \widetilde{R}(u_{enc}), \qquad (11)$$

*where $\widetilde{R}(u)$ is defined as follows:*

$$\widetilde{R}(u) := \frac{2q^2}{K} \sum_{k=1}^{K} (u(\alpha_k) - x_k)^2 + \lambda_e \cdot (m_1 + m_2 M) \left( M + \int_\Omega u''(t)^2 \, dt \right), \qquad (12)$$

*and $m_1, m_2$ are constants.*

*(ii) If $u^*(\cdot)$ is the minimizer of (12), then $\|u^*\|_{\widetilde{\mathcal{H}}^2(\Omega;\mathbb{R})}^2 \leqslant M$.*

See Appendix B.4 for the proof. Proposition 1 states that, under mild assumptions there exists a *smoothing spline* that minimizes the upper bound given in (12) without changing in convergence rate.

**Convergence Rate:** Using Theorems 1, 2, and 3, we can derive the convergence rate of the proposed scheme as stated in the following theorem.

**Theorem 4** (Convergence rate). *For `LeTCC` scheme with $N$ worker nodes and a maximum of $S$ stragglers, $\mathcal{R}(\hat{f}) \leqslant \mathcal{O}\left( S^{\frac{8}{5}} N^{-\frac{3}{5}} \right)$ for the noisy computation, and $\mathcal{R}(\hat{f}) \leqslant \mathcal{O}\left( S^3 N^{-3} \right)$ for the noiseless setting.*

Refer to Appendix B.5 for the proof. Notably, the convergence rate yields from Theorem 4 surpasses the state-of-the-art Berrut coded computing approach upper bound [29] (Figure 1), both with respect to $S$ and $N$ (see Appendix C.1 for detailed comparison).

Table 1: Comparison of the proposed framework (LeTCC) and the state-of-the-art (BACC) in terms of the Root Mean Squared Error (RMSE) and the Relative Accuracy (RelAcc).

| $(N, K, |\mathcal{F}|)$ | LeNet5 (100, 20, 60) | | RepVGG (60, 20, 20) | | ViT (20, 8, 3) | |
|---|---|---|---|---|---|---|
| **Method** | RMSE | RelAcc | RMSE | RelAcc | RMSE | RelAcc |
| BACC | $2.55 \pm 0.43$ | $0.92 \pm 0.04$ | $2.44 \pm 0.38$ | $0.83 \pm 0.05$ | $0.68 \pm 0.13$ | $0.90 \pm 0.07$ |
| LeTCC | $\mathbf{2.18 \pm 0.51}$ | $\mathbf{0.94 \pm 0.04}$ | $\mathbf{2.04 \pm 0.42}$ | $\mathbf{0.87 \pm 0.05}$ | $\mathbf{0.62 \pm 0.11}$ | $\mathbf{0.94 \pm 0.06}$ |

## 5 Experimental Results

In this section, we extensively evaluate the proposed scheme across various scenarios. Our assessments involve examining multiple deep neural networks as computing functions and exploring the impact of different numbers of stragglers on the scheme's efficiency. The experiments are run using PyTorch [35] in a single GPU machine. We evaluate the performance of the LeTCC scheme in three different model architectures:

- **Shallow model**: We choose LeNet5 [36] architecture as a known shallow network with approximately $6 \times 10^4$ parameters, trained on the MNIST [37].

- **Deep model with low-dimensional output**: In this scenario, we evaluate the proposed scheme when the function is a deep neural network trained on color images in CIFAR-10 [38] dataset. We use the recently introduced RepVGG [39] network with around 26 million parameters which was trained on CIFAR-10[1].

- **Deep model with high-dimensional output**: Finally, we demonstrate the performance of the LeTCC scheme in a scenario where the input and output of the computing function are high-dimensional, and the function is a relatively large neural network. We consider the Vision Transformer (ViT) [40] as one of the state-of-the-art base neural networks in computer vision for our prediction model, with more than 80 million parameters (in the base version). The network was trained and fine-tuned on the ImageNet-1K dataset [41][2].

We use the output of the last softmax layer of each model as the output.

**Hyper-parameters:** The entire encoding and decoding process is the same for different functions, as we adhere to a non-parametric approach. The sole hyper-parameters involved are the two smoothing parameters $(\lambda_{\text{enc}}, \lambda_{\text{dec}})$ which are determined using cross-validation and greed search over different values of the smoothing parameters.

**Baseline:** We compare LeTCC with the Berrut approximate coded computing (BACC) introduced by [29] as the state-of-the-art coded computing scheme for general computing. The BACC framework is used in [29] for training neural networks and in [42] for inference. Although Berrut coded computing [29] is the only existing coded computing scheme for general functions, we include a comparison of the proposed framework with the Lagrange coded computing scheme [3] for polynomial computation in Appendix D.

**Interpolation Points:** We choose Chebyshev points of the first and second kind, $\{\alpha_i\}_{k=1}^{K} = \cos\left(\frac{(2k-1)\pi}{2K}\right)$ and $\{\beta_n\}_{n=1}^{N} = \cos\left(\frac{(n-1)\pi}{N-1}\right)$, for fair comparison with [29].

**Evaluation Metrics:** We employ two evaluation metrics to assess the performance of the proposed framework: Relative Accuracy (RelAcc) and Root Mean Squared Error (RMSE). RelAcc is defined as the ratio of the base model's prediction accuracy to the accuracy of the estimated model on the initial data points. RMSE, on the other hand, is our main loss defined in (1) which measures the empirical average of the root mean square difference over multiple batches of and non-straggler set $\mathcal{F}$, providing an unbiased estimation of expected mean square error, $\mathbb{E}_{\mathbf{x} \sim \mathcal{X}, \mathcal{F}}\left[\frac{1}{K} \sum_{k=1}^{K} \|\mathbf{f}(\mathbf{x}_k) - \hat{\mathbf{f}}(\mathbf{x}_k)\|_2\right]$, for data distribution $\mathcal{X}$.

---

[1]The pre-trained weights can be found here.
[2]We use the official PyTorch pre-trained ViT network from here.

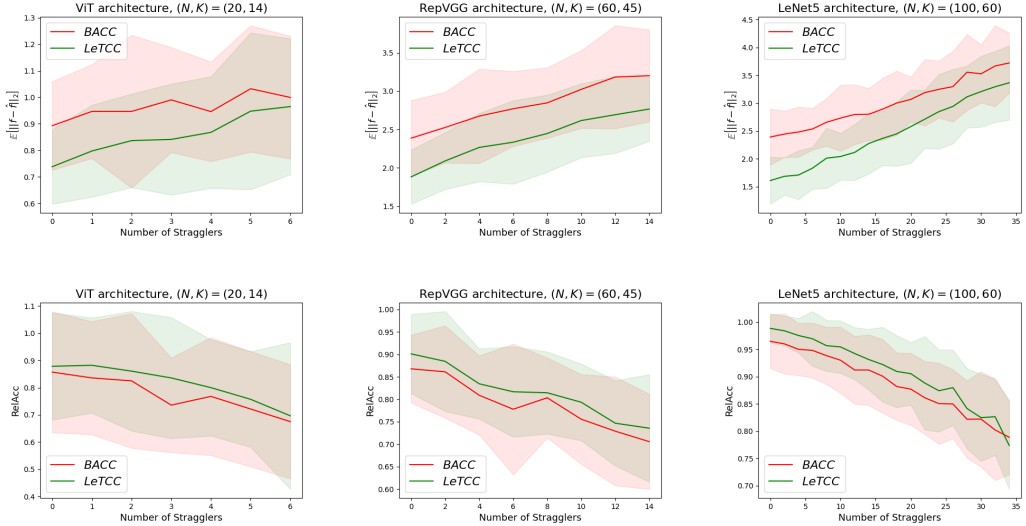

Figure 3: Performance comparison of `LeTCC` and `BACC` with a $95\%$ confidence interval across a diverse range of stragglers for different models in a low-redundancy regime (smaller $\frac{N}{K}$).

**Performance Evaluation:** Table 1 presents both RMSE and RelAcc metrics side by side. The results demonstrate that `LeTCC` outperforms `BACC` across various architectures and configurations, with an average improvement of 15%, 17%, and 9% in RMSE for LeNet, RepVGG, and ViT architectures, respectively, and a 2%, 5%, and 4% enhancement in RelAcc.

In a subsequent analysis, we evaluate the performance of `LeTCC` in comparison to `BACC` across a variety of straggler scenarios. For each number of stragglers, $S$, we randomly select $S$ workers to act as stragglers. Both schemes are then run with the same input data points and straggler configurations, and the process is repeated 20 times. We record the average values of the RelAcc and RMSE metrics, along with their $95\%$ confidence intervals. Figures 3 and 4 illustrate the performance of both schemes across three model architectures. As shown in both figures, the proposed scheme consistently outperforms `BACC` for nearly all straggler values. In Figure 3, where $\frac{N}{K}$ is relatively small–indicating a system design without excessive redundancy, which is more practical–the proposed scheme demonstrates even greater improvements in both metrics.

**Computational Complexity:** The calculation and inference of smoothing spline coefficients can be performed linearly in the number of regression points by leveraging B-spline basis functions [43–45]. Consequently, the encoding and decoding processes in `LeTCC`, which involve evaluating new points and calculating the fitted coefficients, have computational complexities of $\mathcal{O}(K \cdot d)$ and $\mathcal{O}((N - S) \cdot m)$, respectively, where $d$ is the input dimension and $m$ is the output dimension of the computing function $\mathbf{f}(\cdot)$. This complexity is comparable to that of `BACC`, which has complexities of $\mathcal{O}(K)$ and $\mathcal{O}(N - S)$ for its encoding and decoding layers [29]. Table 2 in Appendix C.2 presents a comparison of the total end-to-end processing time statistics for the `LeTCC` and `BACC` schemes.

**Sensitivity Analysis:** We additionally investigate the sensitivity of the proposed schemes performance to the value of the smoothing parameter, as well as the sensitivity of the optimal smoothing parameter to the number of stragglers (or workers). The results are presented in Appendix E. As shown in Table 3 and Figure 6, the optimal smoothing parameters and the scheme's performance exhibit low sensitivity to the number of stragglers (or worker nodes) and to the smoothing parameters, respectively.

**Coded Points:** We also compare the coded points $\{\tilde{\mathbf{x}}_{\mathbf{n}}\}_{n=1}^{N}$ sent to the workers in `LeTCC` and `BACC` schemes. The results, shown in Figure 7, demonstrate that `BACC` coded samples exhibit high-frequency noise which causes the scheme to approximate the original prediction worse than `LeTCC`.

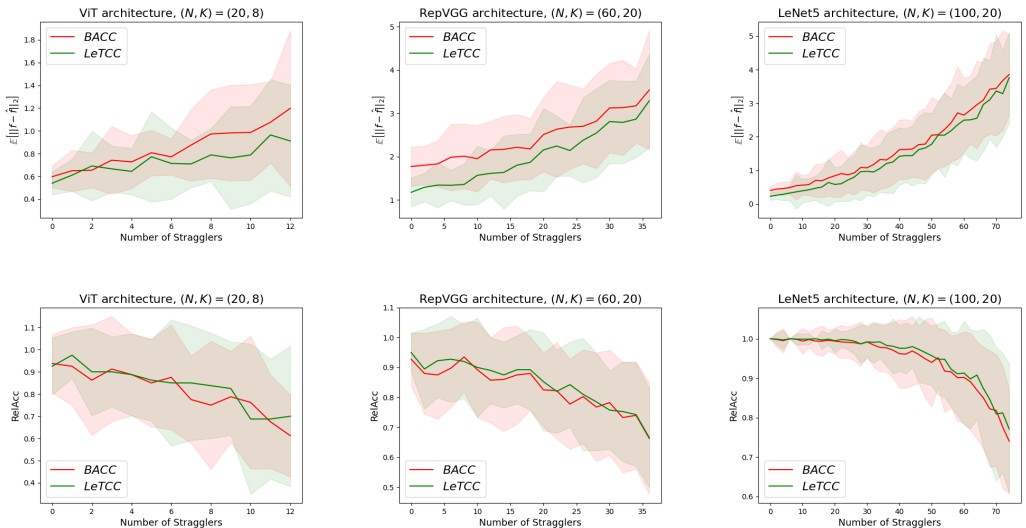

Figure 4: Performance comparison of `LeTCC` and `BACC` with a $95\%$ confidence interval across a diverse range of stragglers for different models in a high-redundancy regime (larger $\frac{N}{K}$).

# 6  Related Work

Coded computing was initially introduced to tackle the challenges of distributed computation, particularly the existence of stragglers or slow workers, and also faulty or adversarial nodes. Traditional approaches to deal with stragglers primarily rely on repetition [1, 46–50], where each task is assigned to multiple workers either proactively or reactively. Recently, coded computing approaches have reduced the overhead of repetition by leveraging coding theory and embedding redundancy in the worker's input data [3, 13, 26, 29, 51–55]. This technique, which mainly relies on theory of coding, has been developed for specific types of structured computations, such as polynomial computation and matrix multiplication [3, 7, 12, 13, 17, 52, 56–58]. Recently, there have been attempts to generalize coded computing for general computations [4, 5, 29, 59]. Towards extending the application of coding computing to machine learning computation, Kosaian et al. [59] suggest training a neural network to predict coded outputs from coded data points. However, the scheme of Kosaian et al. [59] requires a complex training process and tolerates only one straggler. In another work, Jahani-Nezhad and Maddah-Ali [29] proposes `BACC`, a model-agnostic and numerically stable framework for general computations. They successfully employed `BACC` to train neural networks on a cluster of workers, while tolerating a larger number of stragglers. Building on the `BACC` framework, Soleymani et al. [42] introduced `ApproxIFER` scheme, as a straggler resistance and Byzantine-robust prediction serving system. However, the scheme of `BACC` uses a reasonable rational interpolation (Berrut interpolation [60]), off the shelf, for encoding and decoding, without considering any end-to-end cost function to optimize. In contrast, we theoretically formalize a new foundation of coded computing grounded in learning theory, which can be naturally used for machine learning applications.

# 7  Conclusions and Future Work

In this paper, we developed a new foundation for coded computing based on learning theory, contrasting with existing works that rely on coding theory and use metrics like minimum distance and recovery threshold for design. This shift in foundations removes barriers to using coded computing for machine learning applications, allows us to design optimal encoding and decoding functions, and achieves convergence rates that outperform the state of the art. Moreover, the experimental evaluations validate the theoretical guarantees. While this work focuses on straggler mitigation, future work will extend our proposed scheme to achieve Byzantine robustness and privacy, offering promising avenues for further research.

## 8 Acknowledgments

This material is based upon work supported by the National Science Foundation under Grant CIF-2348638. Behrooz Tahmasebi is supported by NSF Award CCF-2112665 (TILOS AI Institute) and NSF Award 2134108.

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

# A  Preliminaries

## A.1  Sobolev spaces and Sobolev norms

Let $\Omega$ be an open interval in $\mathbb{R}$ and $M$ be a positive integer. We denote by $L^p\left(\Omega; \mathbb{R}^M\right)$ the class of all measurable functions $g : \mathbb{R} \to \mathbb{R}^M$ that satisfy:

$$\int_\Omega |g_j(t)|^p \, dt \ < \infty, \quad \forall j \in [M], \tag{13}$$

where $g(\cdot) = [g_1(\cdot), \ldots, g_M(\cdot)]^T$. The space $L^p\left(\Omega; \mathbb{R}^M\right)$ can be endowed with the following norm, known as the $L^p$ norm:

$$\|g\|_{L^p(\Omega;\mathbb{R}^M)} := \left( \sum_{j=1}^M \int_\Omega |g_i(t)|^p \, dt \right)^{\frac{1}{p}}, \tag{14}$$

for $1 \leqslant p < \infty$, and

$$\|g\|_{L^\infty(\Omega;\mathbb{R}^M)} := \max_{j\in[M]} \sup_{t\in\Omega} |g_j(t)|. \tag{15}$$

for $p = \infty$. Additionally, a function $g : \Omega \to \mathbb{R}^M$ is in $L^p_{\mathrm{loc}}\left(\Omega; \mathbb{R}^M\right)$ if it lies in $L^p(V; \mathbb{R}^M)$ for all compact subsets $V \subseteq \Omega$.

**Definition 1** (**Sobolev Space**). *The Sobolev space $\mathbb{W}^{m,p}\left(\Omega; \mathbb{R}^M\right)$ is the space of all functions $g \in L^p\left(\Omega; \mathbb{R}^M\right)$ such that all weak derivatives of order $i$, denoted by $g^{(i)}$, belong to $L^p\left(\Omega; \mathbb{R}^M\right)$ for $i \in [m]$. This space is endowed with the norm:*

$$\|g\|_{\mathbb{W}^{m,p}(\Omega;\mathbb{R}^M)} := \left( \|g\|_{L^p(\Omega;\mathbb{R}^M)}^p + \sum_{i=1}^m \left\| g^{(i)} \right\|_{L^p(\Omega;\mathbb{R}^M)}^p \right)^{\frac{1}{p}}, \tag{16}$$

*for $1 \leqslant p < \infty$, and*

$$\|g\|_{\mathbb{W}^{m,\infty}(\Omega;\mathbb{R}^M)} := \max \left\{ \|g\|_{L^\infty(\Omega;\mathbb{R}^M)}, \max_{i\in[m]} \left\| g^{(i)} \right\|_{L^\infty(\Omega;\mathbb{R}^M)} \right\}, \tag{17}$$

*for $p = \infty$.*

Similarly, $\mathbb{W}^{m,p}_{\mathrm{loc}}\left(\Omega; \mathbb{R}^M\right)$ is defined as the space of all functions $g \in L^p_{\mathrm{loc}}\left(\Omega; \mathbb{R}^M\right)$ with all weak derivatives of order $i$ belonging to $L^p_{\mathrm{loc}}\left(\Omega; \mathbb{R}^M\right)$ for $i \in [m]$. $\|g\|_{\mathbb{W}^{m,p}_{\mathrm{loc}}(\Omega;\mathbb{R}^M)}$ and $\|g\|_{\mathbb{W}^{m,\infty}_{\mathrm{loc}}(\Omega;\mathbb{R}^M)}$ are defined similar to (16) and (19) respectively, using $L^p_{\mathrm{loc}}\left(\Omega; \mathbb{R}\right)$ instead of $L^p\left(\Omega; \mathbb{R}\right)$.

**Definition 2.** (***Sobolev Space with compact support***): *Denoted by $\mathbb{W}^{m,p}_0\left(\Omega; \mathbb{R}^M\right)$ collection of functions $g$ defined on interval $\Omega = (a, b)$ such that $g(a) = \mathbf{0}, g'(a) = \mathbf{0}, \ldots, g^{(m-1)}(a) = \mathbf{0}$ and $\left\| g^{(m)} \right\|_{L^2(\Omega;\mathbb{R})} < \infty$. This space can be endowed with the following norm:*

$$\|g\|_{\mathbb{W}^{m,p}(\Omega;\mathbb{R}^M)} := \left\| g^{(m)} \right\|_{L^p(\Omega;\mathbb{R}^M)}, \tag{18}$$

*for $1 \leqslant p < \infty$, and*

$$\|g\|_{\mathbb{W}^{m,\infty}(\Omega;\mathbb{R}^M)} := \left\| g^{(m)} \right\|_{L^\infty(\Omega;\mathbb{R}^M)}, \tag{19}$$

*for $p = \infty$.*

The next theorem provides an upper bound for $L^p$ norm of functions in the Sobolev space, which plays a crucial role in the proof of the main theorems of the paper.

**Theorem 5** (Theorem 7.34, [61]). *Let $\Omega \subseteq \mathbb{R}$ be an open interval and let $g \in \mathbb{W}^{1,1}_{loc}\left(\Omega; \mathbb{R}^M\right)$. Assume $1 \leqslant p, q, r \leqslant \infty$ and $r \geqslant q$. Then:*

$$\|g\|_{L^r(\Omega;\mathbb{R}^M)} \leqslant \ell^{\frac{1}{r}-\frac{1}{q}} \|g\|_{L^q(\Omega;\mathbb{R}^M)} + \ell^{1-\frac{1}{p}+\frac{1}{r}} \|g'\|_{L^p(\Omega;\mathbb{R}^M)}, \tag{20}$$

*for every $0 < \ell < \mathcal{L}^1(\Omega)$.*

Note that $\mathcal{L}^1(\Omega)$ in Theorem 5 is the length of the interval $\Omega$.

**Corollary 1.** *Suppose $g \in \mathbb{W}_{loc}^{1,1}\left(\Omega; \mathbb{R}^M\right)$ and $\Omega \subseteq \mathbb{R}$ be an open interval. If $\frac{\|g\|_{L^2(\Omega;\mathbb{R}^M)}}{\|g'\|_{L^2(\Omega;\mathbb{R}^M)}} < \mathcal{L}^1(\Omega)$, then:*

$$\|g\|_{L^\infty(\Omega;\mathbb{R}^M)} \leqslant 2\sqrt{\|g\|_{L^2(\Omega;\mathbb{R}^M)} \cdot \|g'\|_{L^2(\Omega;\mathbb{R}^M)}}. \tag{21}$$

*Proof.* Substituting $p, q = 2$ and $r = \infty$ in Theorem 5 and optimizing over $\ell$, one can derive the optimum value of $\ell$, denoted by $\ell^*$ as

$$\ell^* = \frac{\|g\|_{L^2(\Omega;\mathbb{R}^M)}}{\|g'\|_{L^2(\Omega;\mathbb{R}^M)}}. \tag{22}$$

Since $\frac{\|g\|_{L^2(\Omega;\mathbb{R}^M)}}{\|g'\|_{L^2(\Omega;\mathbb{R}^M)}} < \mathcal{L}^1(\Omega)$, the optimum value is in the valid interval mentioned in Theorem 5.
$\square$

**Corollary 2** (Corollary 7.36, [61]). *Let $\Omega = (a, b)$, let $1 \leq p, q, r \leq \infty$ be such that $1 + 1/r \geq 1/p$ and $r \geq q$ and let $g \in \mathbb{W}_{loc}^{1,1}\left(\Omega; \mathbb{R}^M\right)$ with $g' \in L^p\left(\Omega; \mathbb{R}^M\right)$. Let $x_0 \in [a, b]$ be such that $|g(x_0)| = \min_{[a,b]} |g|$. Then*

$$\|g - g(x_0)\|_{L^r(\Omega;\mathbb{R}^M)} \leq 8\|g\|_{L^q(\Omega;\mathbb{R}^M)}^\alpha \|g'\|_{L^p(\Omega;\mathbb{R}^M)}^{1-\alpha} \tag{23}$$

*where $\alpha := 0$ if $r = q$ and $1 - 1/p + 1/r = 0$ and otherwise*

$$\alpha := \frac{1 - 1/p + 1/r}{1 - 1/p + 1/q}.$$

**Corollary 3.** *Theorem 5 and Corollary 2 hold true when $f \in \mathbb{W}^{2,2}\left(\Omega; \mathbb{R}^M\right)$.*

*Proof.* We prove the above corollary by showing that $\mathbb{W}^{2,2}\left(\Omega; \mathbb{R}^M\right) \subseteq \mathbb{W}^{1,1}\left(\Omega; \mathbb{R}^M\right) \subseteq \mathbb{W}_{loc}^{1,1}\left(\Omega; \mathbb{R}^M\right)$. Using Cauchy-Schwartz inequality, one can show:

$$\|f\|_{L^1(\Omega;\mathbb{R}^M)} = \sum_{j=1}^M \int_\Omega |f_j(t)|\, dt \leqslant \sum_{j=1}^M \left(\int_\Omega 1^2\, dt \cdot \int_\Omega |f_j(t)|^2\, dt\right)^{\frac{1}{2}}$$

$$\leqslant \left(\mathcal{L}^1(\Omega)\right)^{\frac{1}{2}} \cdot \sum_{j=1}^M \left(\int_\Omega |f_j(t)|^2\, dt\right)^{\frac{1}{2}} \overset{(a)}{<} \infty, \tag{24}$$

where (a) follows from the bounded length of $\Omega$ and $f \in \mathbb{W}^{2,2}(\Omega; \mathbb{R})$. Similarly:

$$\|f'\|_{L^1(\Omega;\mathbb{R}^M)} = \sum_{j=1}^M \int_\Omega |f_j'(t)|\, dt \leqslant \sum_{j=1}^M \left(\int_\Omega 1^2\, dt \cdot \int_\Omega |f_j'(t)|^2\, dt\right)^{\frac{1}{2}}$$

$$\leqslant \left(\mathcal{L}^1(\Omega)\right)^{\frac{1}{2}} \cdot \sum_{j=1}^M \left(\int_\Omega |f_j'(t)|^2\, dt\right)^{\frac{1}{2}} \overset{(a)}{<} \infty. \tag{25}$$

Equations (24) and (25) prove that $\mathbb{W}^{2,2}\left(\Omega; \mathbb{R}^M\right) \subseteq \mathbb{W}^{1,1}\left(\Omega; \mathbb{R}^M\right)$. Now, suppose $f \in \mathbb{W}^{1,1}\left(\Omega; \mathbb{R}^M\right)$. For every closed subset $[c, d] \subset \Omega$ we have:

$$\sum_{j=1}^M \left(\int_c^d |f_j(t)|\, dt\right) \overset{(a)}{\leqslant} \sum_{j=1}^M \left(\int_\Omega |f_j(t)|\, dt\right) \overset{(b)}{\leqslant} \infty, \tag{26}$$

$$\sum_{j=1}^M \left(\int_c^d |f_j'(t)|\, dt\right) \overset{(a)}{\leqslant} \sum_{j=1}^M \left(\int_\Omega |f_j'(t)|\, dt\right) \overset{(b)}{\leqslant} \infty, \tag{27}$$

where (a) is due to $[c, d] \subset \Omega$ and (b) is because of $f \in \mathbb{W}^{1,1}\left(\Omega; \mathbb{R}^M\right)$. Therefore, $f \in \mathbb{W}_{loc}^{1,1}\left(\Omega; \mathbb{R}^M\right)$.
$\square$

**Equivalent norms.** There have been various norms defined on Sobolev spaces in the literature that are equivalent to (16) (see [62], [63, Ch. 7], and [32, Sec. 10.2]). Note that two norms $\|\cdot\|_{W_1}$, $\|\cdot\|_{W_2}$ are equivalent if there exist positive constants $\eta_1, \eta_2$ such that $\eta_1 \cdot \|g\|_{W_2} \leqslant \|g\|_{W_1} \leqslant \eta_2 \cdot \|g\|_{W_2}$. The equivalent norm in which we are interested is the one introduced in [30]. Let $\Omega = (a, b) \subset \mathbb{R}$. We define $\widetilde{\mathbb{W}}^{m,p}\left(\Omega; \mathbb{R}^M\right)$ as the Sobolev space endowed with the following norm:

$$\|g\|_{\widetilde{\mathbb{W}}^{m,p}(\Omega;\mathbb{R}^M)} := \left( \sum_{j=1}^{M} \left( g_j(a)^p + \sum_{i=1}^{m-1} \left( g_j^{(i)}(a) \right)^p \right) + \left\| g^{(m)} \right\|_{L^p(\Omega;\mathbb{R}^M)}^p \right)^{\frac{1}{p}}. \tag{28}$$

The following lemma derives the equivalence constants $(\eta_1, \eta_2)$ for the norms $\|\cdot\|_{\mathbb{W}^{2,2}(\Omega;\mathbb{R})}$ and $\|\cdot\|_{\widetilde{\mathbb{W}}^{2,2}(\Omega;\mathbb{R})}$.

**Lemma 1.** *Let* $\Omega = (a, b)$ *be an arbitrary open interval in* $\mathbb{R}$. *Then for every* $g \in \mathbb{W}^{2,2}\left(\Omega; \mathbb{R}\right)$:

$$\|g\|_{\mathbb{W}^{2,2}(\Omega;\mathbb{R})}^2 \leqslant \left[ 2(b-a) \max\{1, (b-a)\} \left( 2\max\{1, (b-a)\}^2 + 1 \right) + 1 \right] \cdot \|g\|_{\widetilde{\mathbb{W}}^{2,2}(\Omega;\mathbb{R})}^2 \tag{29}$$

$$\|g\|_{\widetilde{\mathbb{W}}^{2,2}(\Omega;\mathbb{R})}^2 \leqslant \left( \frac{4}{(b-a)} \max\{1, (b-a)\}^2 + 1 \right) \cdot \|g\|_{\mathbb{W}^{2,2}(\Omega;\mathbb{R})}^2. \tag{30}$$

*Proof.* By expanding $g$ around the point $a$ and utilizing the integral remainder form of Taylor's expansion, for every $x \in (a, b)$ we have:

$$g(x) = g(a) + \int_a^x g'(t)\, dt. \tag{31}$$

Therefore:

$$\begin{aligned}
g(x)^2 &= g(a)^2 + \left( \int_a^x g'(t)\, dt \right)^2 + 2g(a) \cdot \int_a^x g'(t)\, dt \\
&\overset{(a)}{\leqslant} 2g(a)^2 + 2 \left( \int_a^x g'(t)\, dt \right)^2 \\
&\overset{(b)}{\leqslant} 2g(a)^2 + 2(x-a) \int_a^x g'(t)^2\, dt \\
&\overset{(c)}{\leqslant} 2g(a)^2 + 2(b-a) \int_a^b g'(t)^2\, dt,
\end{aligned} \tag{32}$$

where (a) follows from the $2g(a) \cdot \int_a^x g'(t)\, dt \leqslant g(a)^2 + \left( \int_a^x g'(t)\, dt \right)^2$, (b) is based on Cauchy-Schwartz inequality, and (c) is due to $x \leqslant b$. Following the same steps as (32), we have:

$$g'(x)^2 \leqslant 2g'(a)^2 + 2(b-a) \int_a^b g''(t)^2\, dt. \tag{33}$$

Integrating both sides of (33) we have:

$$\begin{aligned}
\int_a^b g'(y)^2\, dy &\leqslant 2 \int_a^b g'(a)^2\, dy + 2(b-a) \int_a^b \left( \int_a^b g''(t)^2\, dt \right) dy \\
&= 2(b-a).g'(a)^2 + 2(b-a)^2 \int_a^b g''(t)^2\, dt \\
&\overset{(a)}{\leqslant} 2(b-a) \cdot \max\{1, (b-a)\} \cdot \left( g(a)^2 + g'(a)^2 + \int_a^b g''(t)^2\, dt \right) \\
&= 2(b-a) \cdot \max\{1, (b-a)\} \cdot \|g\|_{\widetilde{\mathbb{W}}^{2,2}(\Omega;\mathbb{R})}^2,
\end{aligned} \tag{34}$$

where (a) is based on adding the positive term $(b-a) \cdot g(a)^2$ to the right-hand side. Based on the (32) and (34) we have:

$$\int_a^b g(x)^2 \, dx \overset{(a)}{\leqslant} 2 \int_a^b g(a)^2 \, dx + 2(b-a) \int_a^b \left( \int_a^b g'(t)^2 \, dt \right) dx$$

$$= 2(b-a).g(a)^2 + 2(b-a)^2 \int_a^b g'(t)^2 \, dt$$

$$\overset{(b)}{\leqslant} 2(b-a) \cdot g(a)^2 + 4(b-a)^3 \cdot g'(a)^2 + 4(b-a)^4 \int_a^b g''(t)^2 \, dt$$

$$\leqslant 4(b-a) \cdot \max\{1, (b-a)^3\} \cdot \|g\|_{\widetilde{\mathbb{W}}^{2,2}(\Omega;\mathbb{R})}^2 , \tag{35}$$

where (a) follows by integrating both sides of (32) and (b) is due to (33). Using (35) and (34) and the fact that $\int_a^b g''(t)^2 \, dt \leqslant \|g\|_{\mathbb{W}^{2,2}(\Omega;\mathbb{R})}^2$, we have:

$$\|g\|_{\mathbb{W}^{2,2}(\Omega;\mathbb{R})}^2 = \int_a^b g''(t)^2 \, dt + \int_a^b g'(t)^2 \, dt + \int_a^b g(t)^2 \, dt$$

$$\leqslant \left[ 2(b-a) \max\{1, (b-a)\} \left( 2 \max\{1, (b-a)\}^2 + 1 \right) + 1 \right] \cdot \|g\|_{\widetilde{\mathbb{W}}^{2,2}(\Omega;\mathbb{R})}^2 . \tag{36}$$

For the other side, using the same steps as in (32), we have:

$$g(a)^2 = g(x)^2 + \left( \int_a^x g'(t) \, dt \right)^2 - 2g(a). \int_a^x g'(t) \, dt$$

$$\overset{(a)}{\leqslant} 2g(x)^2 + 2 \left( \int_a^x g'(t) \, dt \right)^2$$

$$\overset{(b)}{\leqslant} 2g(x)^2 + 2(x-a) \int_a^x g'(t)^2 \, dt$$

$$\overset{(c)}{\leqslant} 2g(x)^2 + 2(b-a) \int_a^b g'(t)^2 \, dt, \tag{37}$$

where (a) is because of $-2g(a) \cdot \int_a^x g'(t) \, dt \leqslant g(a)^2 + \left( \int_a^x g'(t) \, dt \right)^2$, (b) follows from Cauchy-Schwartz inequality, and (c) is due to $x \leqslant b$. Integrating both sides of (37), we have

$$(b-a) \cdot g(a)^2 = \int_a^b g(a)^2 \, dx \leqslant 2 \int_a^b g(x)^2 \, dx + 2(b-a) \int_a^b \left( \int_a^b g'(t)^2 \, dt \right) dx$$

$$\overset{(a)}{=} 2 \int_a^b g(t)^2 \, dt + 2(b-a)^2 \int_a^b g'(t)^2 \, dt$$

$$\overset{(b)}{\leqslant} 2 \max\{1, (b-a)\}^2 \left( \int_a^b g(t)^2 \, dt + \int_a^b g'(t)^2 \, dt + \int_a^b g''(t)^2 \, dt \right)$$

$$\leqslant 2 \max\{1, (b-a)\}^2 \|g\|_{\mathbb{W}^{2,2}(\Omega;\mathbb{R})}^2 , \tag{38}$$

where (a) follows by a change of variable from $x$ to $t$ and (b) follows by adding the positive term $2 \max\{1, (b-a)\}^2 \cdot \int_a^b g''(t)^2 \, dt$ to the right side. Thus, (38) directly results in

$$g(a)^2 \leqslant \frac{2}{(b-a)} \max\{1, (b-a)\}^2 \|g\|_{\mathbb{W}^{2,2}(\Omega;\mathbb{R})}^2 . \tag{39}$$

Following same steps as (38) and (39) results in

$$g'(a)^2 \leqslant \frac{2}{(b-a)} \max\{1, (b-a)\}^2 \|g\|_{\mathbb{W}^{2,2}(\Omega;\mathbb{R})}^2 . \tag{40}$$

Considering the fact that $\int_a^b g''(t)^2 \, dt \leqslant \|g\|_{\mathbb{W}^{2,2}(\Omega;\mathbb{R})}^2$, we can complete the proof:

$$\|g\|_{\widetilde{\mathbb{W}}^{2,2}(\Omega;\mathbb{R})}^2 = g(a)^2 + g'(a)^2 + \int_a^b g''(t)^2 \, dt$$

$$\leqslant \left( \frac{4}{(b-a)} \max\{1, (b-a)\}^2 + 1 \right) \cdot \|g\|_{\mathbb{W}^{2,2}(\Omega;\mathbb{R})}^2 . \tag{41}$$

$\square$

**Corollary 4.** *Based on Lemma 1, for $\Omega = (-1, 1)$ we have*

$$\frac{1}{9} \cdot \|g\|^2_{\widetilde{\mathbb{W}}^{2,2}(\Omega;\mathbb{R})} \leqslant \|g\|^2_{\mathbb{W}^{2,2}(\Omega;\mathbb{R})} \leqslant 73 \cdot \|g\|^2_{\widetilde{\mathbb{W}}^{2,2}(\Omega;\mathbb{R})} . \tag{42}$$

Corollary 4 directly follows from Lemma 1 by substituting $(a, b) = (-1, 1)$.

**Corollary 5.** *The result of Lemma 1 remains valid for multi-dimensional cases, where $g : \mathbb{R} \to \mathbb{R}^M$, for some $M > 1$.*

Corollary 5 directly follows from applying Lemma 1 to each component of the function $g(\cdot)$ and using the definition of vector-valued function norm:

$$\|g\|^2_{\mathbb{W}^{2,2}(\Omega;\mathbb{R}^M)} = \sum_{j=1}^{M} \left\|g^{(j)}\right\|^2_{\mathbb{W}^{2,2}(\Omega;\mathbb{R})}, \quad \|g\|^2_{\widetilde{\mathbb{W}}^{2,2}(\Omega;\mathbb{R}^M)} = \sum_{j=1}^{M} \left\|g^{(j)}\right\|^2_{\widetilde{\mathbb{W}}^{2,2}(\Omega;\mathbb{R})} .$$

**Proposition 2.** *([61, Section 7.2], [63, Theorem 121],[32]) For any open interval $\Omega \subseteq \mathbb{R}$ and $m, M \in \mathbb{N}$,*

$$\begin{aligned}
\mathcal{H}^m\left(\Omega; \mathbb{R}^M\right) &:= \mathbb{W}^{m,2}\left(\Omega; \mathbb{R}^M\right), \\
\widetilde{\mathcal{H}}^m\left(\Omega; \mathbb{R}^M\right) &:= \widetilde{\mathbb{W}}^{m,2}\left(\Omega; \mathbb{R}^M\right), \\
\mathcal{H}_0^m\left(\Omega; \mathbb{R}^M\right) &:= \mathbb{W}_0^{m,2}\left(\Omega; \mathbb{R}^M\right),
\end{aligned} \tag{43}$$

*are Reproducing Kernel Hilbert Spaces (RKHSs).*

The full expression of the kernel function of $\mathcal{H}^m\left(\Omega; \mathbb{R}\right)$ and $\widetilde{\mathcal{H}}^m\left(\Omega; \mathbb{R}\right)$ and other equivalent norms of Sobolev spaces can be found in [63, Section 4]. For $\widetilde{\mathcal{H}}^m\left(\Omega; \mathbb{R}\right)$ the kernel function is as follows:

$$R(t, s) = \sum_{j=0}^{m-1} \frac{t^j s^j}{j!^2} + \int_\Omega \frac{(t-x)_+^{m-1}(s-x)_+^{m-1}}{(m-1)!^2} \, dx, \tag{44}$$

where $(\cdot)_+$ is positive part function.

## A.2 Smoothing Splines

Consider the data model $y_i = f(t_i) + \epsilon_i$ for $i = 1, \dots, n$, where $t_i \in \Omega = (a, b) \subset \mathbb{R}$, $\mathbb{E}[\epsilon_i] = 0$, and $\mathbb{E}[\epsilon_i^2] \leqslant \sigma_0^2$. Assuming $f \in \widetilde{\mathbb{W}}^{m,2}\left(\Omega; \mathbb{R}\right)$, the solution to the following optimization problem is referred to as the smoothing spline:

$$\mathbf{S}_{\lambda,n,m}[\mathbf{y}] := \underset{g \in \widetilde{\mathbb{W}}^{m,2}(\Omega;\mathbb{R})}{\operatorname{argmin}} \frac{1}{n} \sum_{i=1}^{n} \left(g\left(t_i\right) - y_i\right)^2 + \lambda \int_\Omega \left(g^{(m)}(t)\right)^2 \, dt, \tag{45}$$

where $\mathbf{y} = [y_1, \dots, y_n]$. Based on Proposition 2, $\widetilde{\mathcal{H}}^m\left(\Omega; \mathbb{R}\right) := \widetilde{\mathbb{W}}^{m,2}\left(\Omega; \mathbb{R}\right)$ with the norm $\|\cdot\|_{\widetilde{\mathbb{W}}^{m,2}(\Omega;\mathbb{R})}$ is a RKHS for some kernel function $\phi(\cdot, \cdot)$. Therefore for any $v \in \widetilde{\mathbb{W}}^{m,2}\left(\Omega; \mathbb{R}\right)$, we have:

$$v(t) = \langle v(\cdot), \phi(\cdot, t) \rangle_{\widetilde{\mathcal{H}}^m(\Omega;\mathbb{R})}. \tag{46}$$

It can be shown that $\phi(t, s) = R^P(t, s) + R^0(t, s)$ where $R^0(t, s)$ is kernel function of $\mathcal{H}_0^m\left(\Omega; \mathbb{R}\right)$ and $R^P(t, s)$ is a null space of $\mathcal{H}_0^m\left(\Omega; \mathbb{R}\right)$ which is the space of all polynomials with degree less than $m$.

The solution of (45) has the following form [31, 32]:

$$u^*(\cdot) = \sum_{i=1}^{m} d_i \zeta_i(\cdot) + \sum_{j=1}^{n} c_j \nu_j(\cdot), \tag{47}$$

where $\nu_j(\cdot) = R^0(\cdot, t_j)$ for $j \in [n]$, and $\{\zeta_i(\cdot)\}_{i=1}^m$ are the basis functions of the space of polynomials of degree at most $m - 1$. Substituting $u^*$ into (45) and optimizing over $\mathbf{c} = [c_1, \dots, c_n]^T$ and $\mathbf{d} = [d_1, \dots, d_m]^T$, we obtain the following result [32]:

$$\mathbf{S}_{\lambda,n,m}[\mathbf{y}](\mathbf{y}) = \mathbf{Q}\left(\mathbf{Q^T Q} + \lambda \boldsymbol{\Gamma}\right)^{-1} \mathbf{Q^T y}, \tag{48}$$

where

$$\mathbf{Q}_{n \times (n+m)} = \left[\begin{array}{cc} \mathbf{T}_{n \times m} & \boldsymbol{\Sigma}_{n \times n} \end{array}\right],$$

$$\boldsymbol{\Gamma}_{(n+m) \times (n+m)} = \left[\begin{array}{cc} \mathbf{0}_{m \times m} & \mathbf{0}_{m \times n} \\ \mathbf{0}_{n \times 1} & \boldsymbol{\Sigma}_{n \times n} \end{array}\right],$$

$$\boldsymbol{T}_{ij} = \zeta_j(t_i),$$

$$\boldsymbol{\Sigma}_{ij} = R^0(t_i, t_j). \tag{49}$$

Equation (48) states that the smoothing spline fitted on the data points $\mathbf{y}$ is a linear operator:

$$\mathbf{S}_{\lambda,n,m}[\mathbf{y}](\mathbf{z}) := \mathbf{A}_\lambda \mathbf{z}, \tag{50}$$

for $\mathbf{z} \in \mathbb{R}^n$, where $\mathbf{A}_\lambda := \mathbf{Q}\left(\mathbf{Q^T Q} + \lambda \boldsymbol{\Gamma}\right)^{-1} \mathbf{Q^T}$. It can be shown that the $u^*(\cdot)$ is a natural spline [32, 33]. Thus, if $\{b_i(\cdot)\}_{i=1}^n$ is a basis function for an $m$-th order natural spline (such as truncated power or B-spline basis functions), we have:

$$u^*(t) = \sum_{i=1}^n \xi_i b_i(t), \tag{51}$$

$$\boldsymbol{\xi} = \left(\mathbf{N}^T \mathbf{N} + \lambda \Phi\right)^{-1} \mathbf{N}^T \mathbf{y}, \tag{52}$$

where $\mathbf{N}_{ij} = b_i(t_j)$, $\Phi_{ij} = \int_\Omega b_i''(t) b_j''(t)\, dt$ for $i, j \in [n]$, and $\boldsymbol{\xi} := [\xi_1, \dots, \xi_n]^T$.

To characterize the estimation error of the smoothing spline, $|f - \mathbf{S}_{\lambda,n,m}(\mathbf{y})|$, we need to define two variables analogous to those in (6), which quantify the minimum and maximum consecutive distance of the regression points $\{t_i\}_{i=1}^n$:

$$\Delta_{\max} := \max_{i \in \{0\} \cup [n]} \{t_{i+1} - t_i\}, \quad \Delta_{\min} := \min_{i \in [n-1]} \{t_{i+1} - t_i\}, \tag{53}$$

where boundary points are defined as $(t_0, t_{n+1}) := (a, b)$. The following theorem offers an upper bound for the $j$-th derivative of the smoothing spline estimator error function in the absence of noise ($\sigma_0 = 0$).

**Theorem 6.** *([64, Theorem 4.10]) Consider data model $y_i = f(t_i)$ with $\{t_i\}_{i=1}^n$ belong to $\Omega = [a, b]$ for $i \in [n]$. Let*

$$L = p_{2(m-1)}\left(\frac{\Delta_{max}}{\Delta_{min}}\right) \cdot \frac{n \Delta_{max}}{b - a} \frac{\lambda}{2} + D(m) \cdot (\Delta_{max})^{2m}, \tag{54}$$

*where $p_d(\cdot)$ is a degree $d$ polynomial with positive weights and $D(m)$ is a function of $m$. Then for each $j \in \{0, 1, \dots, m\}$ and any $f \in \mathbb{W}^{m,2}(\Omega; \mathbb{R})$, there exist a function $H(m, j)$ such that:*

$$\left\|(f - \mathbf{S}_{\lambda,n,m}(\mathbf{y}))^{(j)}\right\|_{L^2(\Omega;\mathbb{R})}^2 \leqslant H(m, j)\left(1 + \left(\frac{L}{(b-a)^{2m}}\right)\right)^{\frac{j}{m}} \cdot L^{\frac{(m-j)}{m}} \cdot \left\|f^{(m)}\right\|_{L^2(\Omega;\mathbb{R})}^2. \tag{55}$$

Note that $(f - \mathbf{S}_{\lambda,n,m}(\mathbf{y}))^{(0)} := f - \mathbf{S}_{\lambda,n,m}(\mathbf{y})$. In the presence of noise, where $\sigma_0 > 0$, we can exploit the linearity of the smoothing spline operator and the mutual independence of the noise terms to conclude that:

$$\mathbb{E}_\epsilon\left[\|(f - \mathbf{S}_{\lambda,n,m}(\mathbf{y}))\|_{L^2(\Omega;\mathbb{R})}^2\right] = \mathbb{E}_\epsilon\left[\|(f - \mathbf{S}_{\lambda,n,m}[\mathbf{y}](\mathbf{f} + \boldsymbol{\epsilon}))\|_{L^2(\Omega;\mathbb{R})}^2\right]$$

$$= \|(f - \mathbf{S}_{\lambda,n,m}[\mathbf{y}](\mathbf{f}))\|_{L^2(\Omega;\mathbb{R})}^2$$

$$+ \mathbb{E}_\epsilon\left[\|(f - \mathbf{S}_{\lambda,n,m}[\mathbf{y}](\boldsymbol{\epsilon}))\|_{L^2(\Omega;\mathbb{R})}^2\right], \tag{56}$$

where $\mathbf{f} = [f(t_1), \dots, f(t_n)]^T$. The first term in (56) can be upper bounded using Theorem 6. The following theorem establishes an upper bound for the second term when $\frac{\Delta_{\max}}{\Delta_{\min}}$ is bounded:

**Theorem 7.** *([65, Section 5]) Consider data model $y_i = f(t_i) + \epsilon_i$, where $\mathbb{E}[\epsilon_i] = 0$ and $\mathbb{E}[\epsilon_i^2] \leqslant \sigma_0^2$ for $i \in [n]$. Assume there exist a constant $B > 0$ such that $\frac{\Delta_{max}}{\Delta_{min}} \leqslant B$. Then for each $j \in \{0, 1, \ldots, m\}$, there exist a constant $\lambda_0 > 0$ and function $Q(m, j, \lambda_0)$ such that:*

$$\mathbb{E}_{\boldsymbol{\epsilon}}\left[\left\|(f - \mathbf{S}_{\lambda,n,m}[\mathbf{y}](\boldsymbol{\epsilon}))^{(j)}\right\|^2_{L^2(\Omega;\mathbb{R})}\right] \leqslant \frac{Q(m, j, \lambda_0) \cdot \sigma_0^2}{n} \lambda^{\frac{-(2j+1)}{2m}} \tag{57}$$

*for $\lambda \leqslant \lambda_0$ and $n\lambda^{\frac{1}{2m}} \geqslant 1$.*

Note that based on [65], $Q(m, j, \lambda_0) = w(m, j)\lambda_0^{\frac{1}{2m}} + \tilde{w}(m, j)$.

# B    Proof of Theorems

Recall from (2) that $\mathcal{R}(\hat{f}) \leqslant \mathcal{L}_{\text{enc}}(\hat{f}) + \mathcal{L}_{\text{dec}}(\hat{f})$, where

$$\mathcal{L}_{\text{dec}}(\hat{f}) = \mathbb{E}_{\epsilon, \mathcal{F} \sim F_{S,N}}\left[\frac{2}{K}\sum_{k=1}^{K}(u_{\text{dec}}(\alpha_k) - f(u_{\text{enc}}(\alpha_k)))^2\right], \tag{58}$$

$$\mathcal{L}_{\text{enc}}(\hat{f}) = \frac{2}{K}\sum_{k=1}^{K}(f(u_{\text{enc}}(\alpha_k)) - f(x_k))^2. \tag{59}$$

We begin by deriving a general intermediate bound for $\mathcal{L}_{\text{dec}}$ and $\mathcal{L}_{\text{enc}}$ which will be a key component in the proofs of Theorems 2 and 1. The subsequent subsections will then provide the remaining details to complete the proofs of both theorems.

**Lemma 2.** *Let $f : \mathbb{R} \to \mathbb{R}$ be a q-Lipschitz continuous function. Then:*

$$\mathcal{L}_{enc} \leqslant \frac{2q^2}{K}\sum_{k=1}^{K}(u_{enc}(\alpha_k) - x_k)^2. \tag{60}$$

*Proof.* Using Lipschitz property, we have:

$$\begin{aligned}
\mathcal{L}_{\text{enc}} &= \frac{2}{K}\sum_{k=1}^{K}(f(u_{\text{enc}}(\alpha_k)) - f(x_k))^2 \\
&= \frac{2}{K}\sum_{k=1}^{K}(|f(u_{\text{enc}}(\alpha_k)) - f(x_k)|)^2 \\
&\leqslant \frac{2}{K}\sum_{k=1}^{K}(q \cdot |u_{\text{enc}}(\alpha_k) - x_k|)^2 \\
&= \frac{2q^2}{K}\sum_{k=1}^{K}(u_{\text{enc}}(\alpha_k) - x_k)^2.
\end{aligned} \tag{61}$$

$\square$

As previously mentioned, $\mathcal{L}_{\text{dec}}$ represents the expected generalization error of the decoder function. To leverage the results from Theorems 6 and 7 we must establish that the composition of $f$ with the encoder $u_{\text{enc}}$ belongs to the Sobolev space $\mathbb{W}^{2,2}(\Omega;\mathbb{R})$.

**Lemma 3.** *Let $f : \mathbb{R} \to \mathbb{R}$ be a q-Lipschitz continuous function with $\|f''\|_{L^\infty(\Omega;\mathbb{R})} \leqslant \nu$ and $\Omega \subset \mathbb{R}$ be an open interval. If $u_{enc} \in \mathbb{W}^{2,2}(\Omega;\mathbb{R})$ then $f \circ u_{enc} \in \mathbb{W}^{2,2}(\Omega;\mathbb{R})$.*

*Proof.* Let us define $f_0(t) := f(t) - f(0)$. Thus $f_0(0) = 0$ and $f_0$ is $q$-Lipschitz. Using Lipschitz property

$$|f_0(u_{\text{enc}}(t))|^2 = |f_0(u_{\text{enc}}(t)) - f_0(0)|^2 \leqslant q^2 \cdot u_{\text{enc}}(t)^2. \tag{62}$$

Integrating both sides of (62):

$$\int_\Omega (f_0 \circ u_{\text{enc}}(t))^2 \, dt \leqslant q^2 \cdot \int_\Omega u_{\text{enc}}(t)^2 \, dt \overset{(a)}{<} \infty, \tag{63}$$

where (a) follows by $u_{\text{enc}} \in \mathbb{W}^{2,2}(\Omega; \mathbb{R})$. Given that $f_0$ is $q$-Lipschitz, its derivative is bounded in the $L^\infty(\Omega; \mathbb{R})$-norm, i.e., $\|f_0'\|_{L^\infty(\Omega;\mathbb{R})} \leqslant q$. Thus

$$\int_\Omega ((f_0 \circ u_{\text{enc}}(t))')^2 \, dt \overset{(a)}{=} \int_\Omega (f_0'(u_{\text{enc}}(t)))^2 \cdot u_{\text{enc}}'(t)^2 \, dt \leqslant q^2 \cdot \int_\Omega u_{\text{enc}}'(t)^2 \, dt \overset{(b)}{<} \infty, \tag{64}$$

where (a) follows by chain rule and (b) follows by $u_{\text{enc}} \in \mathbb{W}^{2,2}(\Omega; \mathbb{R})$. For the second derivative we have:

$$\begin{aligned}
\int_\Omega \left[ f_0(u_{\text{enc}}(t)'')\right]^2 \, dt &\overset{(a)}{=} \int_\Omega \left[ u_{\text{enc}}''(t) \cdot f_0'(u_{\text{enc}}(t)) + u_{\text{enc}}'(t)^2 \cdot f_0''(u_{\text{enc}}(t)) \right]^2 \, dt \\
&\overset{(b)}{\leqslant} \int_\Omega \left[ u_{\text{enc}}''(t)^2 + u_{\text{enc}}'(t)^4 \right] \left[ f_0'(u_{\text{enc}}(t))^2 + f_0''(u_{\text{enc}}(t))^2 \right] \, dt \\
&\overset{(c)}{\leqslant} (q^2 + \nu^2) \int_\Omega \left[ u_{\text{enc}}''(t)^2 + u_{\text{enc}}'(t)^4 \right] \, dt \\
&= (q^2 + \nu^2) \left( \|u_{\text{enc}}''(t)\|_{L^2(\Omega;\mathbb{R})}^2 + \|u_{\text{enc}}'(t)\|_{L^4(\Omega;\mathbb{R})}^4 \right) \\
&\overset{(d)}{\leqslant} (q^2 + \nu^2) \left( \|u_{\text{enc}}''(t)\|_{L^2(\Omega;\mathbb{R})}^2 + \left( \|u_{\text{enc}}'(t)\|_{L^2(\Omega;\mathbb{R})} + \|u_{\text{enc}}''(t)\|_{L^2(\Omega;\mathbb{R})} \right)^4 \right) \\
&\overset{(e)}{<} \infty,
\end{aligned} \tag{65}$$

where (a) follows from the chain rule, (b) is derived using the Cauchy-Schwartz inequality, (c) is due to $\|f_0''\|_{L^\infty(\Omega;\mathbb{R})} \leqslant \nu$, (d) follows from Theorem 5 with $r = 4, p = q = 2, l = 1$, and (e) is a result of $u_{\text{enc}} \in \mathbb{W}^{2,2}(\Omega; \mathbb{R})$. Equations (63), (64), and (63) demonstrate that $f_0 \circ u_{\text{enc}} \in \mathbb{W}^{2,2}(\Omega; \mathbb{R})$. Note that $\Omega$ is bounded, and every constant function belongs to $\mathbb{W}^{2,2}(\Omega; \mathbb{R})$. Thus, we can conclude that $f_0 \circ u_{\text{enc}}(t) + f(0) = f \circ u_{\text{enc}}(t) \in \mathbb{W}^{2,2}(\Omega; \mathbb{R})$. $\qquad\square$

Let us define the function $h(t) := u_{\text{dec}}(t) - f(u_{\text{enc}}(t))$. Based on Lemma 3, and given that $u_{\text{dec}}$ and $f \circ u_{\text{enc}}$ belong to the Sobolev space $\mathbb{W}^{2,2}(\Omega; \mathbb{R})$, it follows that $h \in \mathbb{W}^{2,2}(\Omega; \mathbb{R})$. The subsequent lemmas establish upper bounds for $\|h\|_{L^\infty(\Omega;\mathbb{R})}$ and $\|h'\|_{L^\infty(\Omega;\mathbb{R})}$, leveraging properties of functions in Sobolev spaces.

**Lemma 4.** *If $\Omega = (-a, a)$, then:*

$$\|h\|_{L^2(\Omega;\mathbb{R})} \leqslant \|h'\|_{L^2(\Omega;\mathbb{R})} \cdot \sqrt{x_0^2 + a^2} \leqslant \|h'\|_{L^2(\Omega;\mathbb{R})} \cdot \sqrt{2}a \tag{66}$$

*Proof.* Assume $\exists x_0 \in \Omega : h(x_0) = 0$. Therefore, $|h(x)| = \left| \int_{x_0}^x h'(x) \, dx \right|$ for $x \in [x_0, a)$. Thus

$$|h(x)| = \left| \int_{x_0}^x h'(x) \, dx \right| \overset{(a)}{\leqslant} \int_{x_0}^x |h'(x)| \, dx \overset{(b)}{\leqslant} \left( \int_{x_0}^x 1^2 \, dx \right)^{\frac{1}{2}} \left( \int_{x_0}^x |h'(x)|^2 \, dx \right)^{\frac{1}{2}}, \tag{67}$$

where (a) and (b) are followed by the triangle and Cauchy-Schwartz inequalities respectively. Integrating the square of both sides over the interval $[x_0, a)$ yields:

$$\int_{x_0}^a |h(x)|^2 \, dx \leqslant \int_{x_0}^a (z - x_0) \cdot \left( \int_{x_0}^a |h'(x)|^2 \, dx \right) \, dz \overset{(a)}{\leqslant} \int_{x_0}^a (z - x_0) \, dz \cdot \int_{-a}^a |h'(x)|^2 \, dx, \tag{68}$$

where (a) follows by $x_0 < a$. On the other side, we have the following for every $x \in (-a, x_0]$:

$$|h(x)| = \left| \int_x^{x_0} h'(x) \, dx \right| \leqslant \int_x^{x_0} |h'(x)| \, dx \leqslant \left( \int_x^{x_0} 1^2 \, dx \right)^{\frac{1}{2}} \cdot \left( \int_x^{x_0} |h'(x)|^2 \, dx \right)^{\frac{1}{2}}. \tag{69}$$

Therefore, we have a similar inequality:

$$\int_{-a}^{x_0} |h(x)|^2 \, dx \leqslant \int_{-a}^{x_0} (x_0 - x) \, dx \cdot \int_{-a}^{a} |h'(x)|^2 \, dx. \tag{70}$$

Using (68) and (70) completes the proof:

$$\|h\|_{L^2(\Omega)}^2 = \int_{-a}^{x_0} |h(x)|^2 \, dx + \int_{x_0}^{a} |h(x)|^2 \, dx$$

$$\leqslant \|h'\|_{L^2(\Omega;\mathbb{R})}^2 \cdot \left( \int_{-a}^{x_0} (x_0 - x) \, dx + \int_{x_0}^{a} (x - x_0) \, dx \right)$$

$$\leqslant \|h'\|_{L^2(\Omega;\mathbb{R})}^2 \cdot \left( x_0 (x_0 + a) - \left( \frac{x_0^2 - a^2}{2} \right) + \left( \frac{a^2 - x_0^2}{2} \right) - x_0 (a - x_0) \right) \tag{71}$$

$$= \|h'\|_{L^2(\Omega;\mathbb{R})}^2 \cdot (x_0^2 + a^2) \leqslant \|h'\|_{L^2(\Omega;\mathbb{R})}^2 \, 2a^2. \tag{72}$$

Thus, if $x_0$ exists, the proof is complete. In the next step, we prove the existence of $x_0 \in \Omega$ such that $h(x_0) = 0$. Recall that $h(t) = u_{\text{dec}}(t) - f(u_{\text{enc}}(t))$ and $u_{\text{dec}}(\cdot)$ is the solution of (4). Assume there is no such $x_0$. Since $h \in \mathbb{W}^{2,2}(\Omega;\mathbb{R})$, then $h(\cdot)$ is continuous. Therefore, if there exist $t_1, t_2 \in \Omega$ such that $h(t_1) < 0$ and $h(t_2) > 0$, then the intermediate value theorem states that there exists $x_0 \in (t_1, t_2)$ such that $h(x_0) = 0$. Thus, $h(t) > 0$ or $h(t) < 0$ for all $t \in \Omega$. Without loss of generality, assume the first case where $h(t) > 0$ for all $t \in \Omega$. It means that $u_{\text{dec}}(t) > f(u_{\text{enc}}(t))$ for all $t \in \Omega$. Let us define

$$\beta^* := \underset{\beta \in \mathcal{F}}{\arg\min} \, u_{\text{dec}}(\beta) - f(u_{\text{enc}}(\beta)).$$

Let $\bar{u}_{\text{dec}}(t) := u_{\text{dec}}(t) - u_{\text{dec}}(\beta^*)$. Note that $\int_\Omega (\bar{u}_{\text{dec}}''(t))^2 \, dt = \int_\Omega (u_{\text{dec}}''(t))^2 \, dt$. Therefore,

$$\sum_{v \in \mathcal{F}} [u_{\text{dec}}(\beta_v) - f(u_{\text{enc}}(\beta_v))]^2 = \sum_{v \in \mathcal{F}} [\bar{u}_{\text{dec}}(\beta_v) + u_{\text{dec}}(\beta^*) - f(u_{\text{enc}}(\beta_v))]^2$$

$$= \sum_{v \in \mathcal{F}} [\bar{u}_{\text{dec}}(\beta_v) - f(u_{\text{enc}}(\beta_v))]^2 + |\mathcal{F}| \cdot u_{\text{dec}}(\beta^*)^2$$

$$+ 2 u_{\text{dec}}(\beta^*) \sum_{v \in \mathcal{F}} [\bar{u}_{\text{dec}}(\beta_v) - f(u_{\text{enc}}(\beta_v))]$$

$$\overset{(a)}{\geqslant} \sum_{v \in \mathcal{F}} [\bar{u}_{\text{dec}}(\beta_v) - f(u_{\text{enc}}(\beta_v))]^2, \tag{73}$$

where (a) follows from $u_{\text{dec}}(\beta^*) > 0$ and $\bar{u}_{\text{dec}}(\beta_v) > f(u_{\text{enc}}(\beta_v))$ for all $v \in \mathcal{F}$. This leads to a contradiction since it implies that $u_{\text{dec}}$ is not the solution of the (4). Therefore, our initial assumption must be wrong. Thus, there exists $x_0 \in \Omega$ such that $h(x_0) = 0$. $\qquad \square$

**Lemma 5.** *Let* $\Omega = (-1, 1)$. *For* $h(t) = u_{dec}(t) - f(u_{enc}(t))$ *we have:*

$$\|h\|_{L^\infty(\Omega;\mathbb{R})} \leqslant 2 \|h\|_{L^2(\Omega;\mathbb{R})}^{\frac{1}{2}} \cdot \|h'\|_{L^2(\Omega;\mathbb{R})}^{\frac{1}{2}} < \infty, \tag{74}$$

*and*

$$\|h'\|_{L^\infty(\Omega;\mathbb{R})} \leqslant \|h'\|_{L^2(\Omega;\mathbb{R})} + \|h''\|_{L^2(\Omega;\mathbb{R})} < \infty. \tag{75}$$

*Proof.* Using Lemma 4 one can conclude

$$\frac{\|h\|_{L^2(\Omega;\mathbb{R})}}{\|h'\|_{L^2(\Omega;\mathbb{R})}} \leqslant \sqrt{2}. \tag{76}$$

Since $h \in \mathbb{W}^{2,2}(\Omega;\mathbb{R})$ we can apply Corollary 1 and Theorem 5 with $r = \infty$ and $p, q = 2$ to complete the proof of (74). Furthermore, using Theorem 5 with $r = \infty$ and $p, q = 2$ and $\ell = 1$ completes the proof of (75). $\qquad \square$

Building upon Lemma 5 and starting from (58), we can derive an upper bound for $\mathcal{L}_{\text{dec}}$:

$$\mathcal{L}_{\text{dec}}(u_{\text{dec}}) = \mathbb{E}_{\epsilon,\mathcal{F}}\left[\frac{2}{K}\sum_{k=1}^{K}(u_{\text{dec}}(\alpha_k) - f(u_{\text{enc}}(\alpha_k)))_2^2\right]$$

$$\stackrel{(a)}{=} \frac{2}{K}\sum_{k=1}^{K}\mathbb{E}_{\epsilon,\mathcal{F}}\left[h(\alpha_k)^2\right]$$

$$\stackrel{(b)}{\leqslant} \frac{2}{K}\sum_{k=1}^{K}\mathbb{E}_{\epsilon,\mathcal{F}}\left[\|h\|_{L^\infty(\Omega;\mathbb{R})}^2\right]$$

$$\stackrel{(c)}{\leqslant} 2\mathbb{E}_{\epsilon,\mathcal{F}}\left[\|h\|_{L^2(\Omega;\mathbb{R})}\cdot\|h'\|_{L^2(\Omega;\mathbb{R})}\right]$$

$$\stackrel{(d)}{\leqslant} 2\mathbb{E}_{\epsilon,\mathcal{F}}\left[\|h\|_{L^2(\Omega;\mathbb{R})}^2\right]^{\frac{1}{2}}\cdot\mathbb{E}_{\epsilon,\mathcal{F}}\left[\|h'\|_{L^2(\Omega;\mathbb{R})}^2\right]^{\frac{1}{2}}, \tag{77}$$

where (a) follows from the definition of $h(t)$, (b) is due to the fact that $h(t) \leqslant \|h\|_{L^\infty(\Omega;\mathbb{R})}$ for $t \in \Omega$, (c) follows by applying Lemma 5, and (d) is a result of applying the Cauchy-Schwartz inequality.

## B.1   Proof of Theorem 1

*Proof.* As previously mentioned, $u_{\text{dec}}(\cdot)$ is a second-order smoothing spline fitted on the data points $\left\{(\beta_{i_1}, f(u_{\text{enc}}(\beta_{i_1}))),\ldots,(\beta_{i_{|\mathcal{F}|}}, f(u_{\text{enc}}(\beta_{i_{|\mathcal{F}|}})))\right\}$, where $\mathcal{F} := \left\{\beta_{i_1},\ldots,\beta_{i_{|\mathcal{F}|}}\right\}$ represents the set of non-straggler worker nodes, and $\boldsymbol{f}\circ\boldsymbol{u}_{\text{enc}} := \left\{f(u_{\text{enc}}(\beta_{i_1})),\ldots,f(u_{\text{enc}}(\beta_{i_{|\mathcal{F}|}}))\right\}$ is the corresponding set of computation results from these non-straggler workers. By the definition given in (50), $\mathbf{S}_{\lambda_{\text{d}},|\mathcal{F}|,2}(\cdot)$ denotes the smoothing spline operator for the decoder layer. Hence, we have the following:

$$\mathbb{E}_{\mathcal{F}}\left[\|h\|_{L^2(\Omega;\mathbb{R})}^2\right] = \mathbb{E}_{\mathcal{F}}\left[\left\|f\circ u_{\text{enc}} - \mathbf{S}_{\lambda_{\text{d}},|\mathcal{F}|,2}[\mathbf{f}]\right\|_{L^2(\Omega;\mathbb{R})}^2\right], \tag{78}$$

where $\mathbf{f} = \left\{f(u_{\text{enc}}(\beta_{i_1})),\ldots,f(u_{\text{enc}}(\beta_{i_{|\mathcal{F}|}}))\right\}$. Let us define the following variables analogous to those in (6):

$$\Delta_{\max}^{\mathcal{F}} := \max_{f\in\{0,\ldots,|\mathcal{F}|\}}\left\{\beta_{i_{f+1}} - \beta_{i_f}\right\}, \quad \Delta_{\min}^{\mathcal{F}} := \min_{f\in\{1,\ldots,|\mathcal{F}|-1\}}\left\{\beta_{i_{f+1}} - \beta_{i_f}\right\}. \tag{79}$$

Since there are at most $S$ stragglers among the worker nodes, we have $\Delta_{\max}^{\mathcal{F}} \leqslant (S+1)\cdot\Delta_{\max}$ and $\frac{\Delta_{\max}^{\mathcal{F}}}{\Delta_{\min}^{\mathcal{F}}} \leqslant (S+1)\cdot\frac{\Delta_{\max}}{\Delta_{\min}} \leqslant (S+1)B$. Additionally, because $\Delta_{\min} \leqslant \frac{2}{N}$, there exists a constant $J$ such that $\Delta_{\max} \leqslant \frac{J}{N}$, and consequently, $\Delta_{\max}^{\mathcal{F}} \leqslant \frac{J(S+1)}{N} \leqslant \frac{J(S+1)}{N-S}$. Otherwise, this would contradict the condition $\frac{\Delta_{\max}}{\Delta_{\min}} \leqslant B$.

Applying Theorem 6 with $\Omega = (-1,1), m = 2$, we have

$$\mathbb{E}_{\mathcal{F}}\left[\|h\|_{L^2(\Omega;\mathbb{R})}^2\right] \leqslant H_0\left\|(f\circ u_{\text{enc}})^{(2)}\right\|_{L^2(\Omega;\mathbb{R})}^2\cdot\mathbb{E}_{\mathcal{F}}[L], \tag{80}$$

and

$$\mathbb{E}_{\mathcal{F}}\left[\|h'\|_{L^2(\Omega;\mathbb{R})}^2\right] \leqslant H_1\left\|(f\circ u_{\text{enc}})^{(2)}\right\|_{L^2(\Omega;\mathbb{R})}^2\cdot\mathbb{E}_{\mathcal{F}}\left[L^{\frac{1}{2}}(1+\frac{L}{16})^{\frac{1}{2}}\right], \tag{81}$$

where $L = p_2\left(\frac{\Delta_{\max}^{\mathcal{F}}}{\Delta_{\min}^{\mathcal{F}}}\right)\cdot\frac{(N-S)\Delta_{\max}^{\mathcal{F}}}{4}\lambda_{\text{d}} + D(2)\cdot\left(\Delta_{\max}^{\mathcal{F}}\right)^4$ and $H_0, H_1 := H(2,0), H(2,1)$ as defined in Theorem 6. Thus, we have:

$$\mathbb{E}_{\mathcal{F}}[L] \leqslant \mathbb{E}_{\mathcal{F}}\left[p_2\left(\frac{\Delta_{\max}^{\mathcal{F}}}{\Delta_{\min}^{\mathcal{F}}}\right)\cdot\frac{(N-S)\Delta_{\max}^{\mathcal{F}}}{4}\lambda_{\text{d}} + D\cdot\left(\Delta_{\max}^{\mathcal{F}}\right)^4\right]$$

$$\stackrel{(a)}{\leqslant} \mathbb{E}_{\mathcal{F}}\left[p_2\left(\frac{\Delta_{\max}^{\mathcal{F}}}{\Delta_{\min}^{\mathcal{F}}}\right)\right]\cdot\frac{J(S+1)}{4}\lambda_{\text{d}} + D\cdot\mathbb{E}_{\mathcal{F}}\left[\left(\Delta_{\max}^{\mathcal{F}}\right)^4\right] \tag{82}$$

$$\stackrel{(b)}{\leqslant} \mathbb{E}_{\mathcal{F}}\left[p_2\left(\frac{\Delta_{\max}^{\mathcal{F}}}{\Delta_{\min}^{\mathcal{F}}}\right)\right]\cdot\frac{J(S+1)}{4}\lambda_{\text{d}} + DJ^4\frac{(S+1)^4}{N^4}, \tag{83}$$

where $D := D(2)$ and (a) and (b) follow from $\Delta_{\max}^{\mathcal{F}} \leqslant \frac{J(S+1)}{N-S}$ and $\Delta_{\max}^{\mathcal{F}} \leqslant \frac{J(S+1)}{N}$ respectively. Substituting in (91),(92), and (92), we have:

$$\mathbb{E}_{\mathcal{F}}\left[\|h\|_{L^2(\Omega;\mathbb{R})}^2\right] \leqslant H_0 \left\|(f \circ u_{\text{enc}})^{(2)}\right\|_{L^2(\Omega;\mathbb{R})}^2 \cdot \left(\mathbb{E}_{\mathcal{F}}\left[p_2\left(\frac{\Delta_{\max}^{\mathcal{F}}}{\Delta_{\min}^{\mathcal{F}}}\right)\right] \cdot \frac{J(S+1)}{4}\lambda_{\text{d}} + DJ^4\frac{(S+1)^4}{N^4}\right),$$
(84)

$$\mathbb{E}_{\mathcal{F}}\left[\|h'\|_{L^2(\Omega;\mathbb{R})}^2\right] \leqslant H_1 \left\|(f \circ u_{\text{enc}})^{(2)}\right\|_{L^2(\Omega;\mathbb{R})}^2 \cdot \left(\mathbb{E}_{\mathcal{F}}\left[p_2\left(\frac{\Delta_{\max}^{\mathcal{F}}}{\Delta_{\min}^{\mathcal{F}}}\right)\right] \cdot \frac{J(S+1)}{4}\lambda_{\text{d}} + DJ^4\frac{(S+1)^4}{N^4}\right)^{\frac{1}{2}}$$
$$\cdot \left(1 + \frac{\mathbb{E}_{\mathcal{F}}\left[p_2\left(\frac{\Delta_{\max}^{\mathcal{F}}}{\Delta_{\min}^{\mathcal{F}}}\right)\right] \cdot \frac{J(S+1)}{4}\lambda_{\text{d}} + DJ^4\frac{(S+1)^4}{N^4}}{16}\right)^{\frac{1}{2}}.$$
(85)

Therefore, we can derive an upper bound for (77) based on the above inequalities. This upper bound holds for any $\lambda_{\text{d}}$. Since $\lambda_{\text{d}} \leqslant N^{-4}$ and $\frac{\Delta_{\max}^{\mathcal{F}}}{\Delta_{\min}^{\mathcal{F}}} \leqslant (S+1)B$, we have:

$$\mathbb{E}_{\mathcal{F}}\left[p_2\left(\frac{\Delta_{\max}^{\mathcal{F}}}{\Delta_{\min}^{\mathcal{F}}}\right)\right] \cdot \frac{J(S+1)}{4}\lambda_{\text{d}} + DJ^4\frac{(S+1)^4}{N^4} \leqslant \tilde{p}_3(S+1)N^{-4} + DJ^4\frac{(S+1)^4}{N^4}$$
$$\overset{(a)}{\leqslant} \widetilde{D}\frac{(S+1)^4}{N^4},$$
(86)

where $\tilde{p}_3$ is a degree-3 polynomial in $(S+1)$ with positive constant coefficients, and $\widetilde{D}$ is the sum of the coefficients of $\tilde{p}_3$ and $DJ^4$. Therefore, we have:

$$\mathcal{L}_{\text{dec}} \leqslant \left\|(f \circ u_{\text{enc}})^{(2)}\right\|_{L^2(\Omega;\mathbb{R})}^2 \left[(H_0 \cdot r(S,N))^{\frac{1}{2}}\left(H_1 \cdot r(S,N)^{\frac{1}{2}}\left(1 + \frac{r(S,N)^{\frac{1}{2}}}{16}\right)\right)^{\frac{1}{2}}\right]$$
$$= \left\|(f \circ u_{\text{enc}})^{(2)}\right\|_{L^2(\Omega;\mathbb{R})}^2 \left[H_0^{\frac{1}{2}}H_1^{\frac{1}{2}} \cdot r(S,N)^{\frac{3}{4}} \cdot \left(1 + \frac{r(S,N)}{16}\right)^{\frac{1}{4}}\right],$$
(87)

where $r(S,N) := \widetilde{D}\frac{(S+1)^4}{N^4}$. Note that, since $S+1 \leqslant N$ then $1 + \frac{r(S,N)}{16} \leqslant 2\max(1,\widetilde{D})$. Defining $\eta := 2\max(1,\widetilde{D})$, we have:

$$\mathcal{L}_{\text{dec}} \leqslant \left\|(f \circ u_{\text{enc}})^{(2)}\right\|_{L^2(\Omega;\mathbb{R})}^2 \left[H_0^{\frac{1}{2}}H_1^{\frac{1}{2}}\eta^{\frac{1}{4}} \cdot r(S,N)^{\frac{3}{4}}\right],$$
(88)

Defining $C := H_0^{\frac{1}{2}}H_1^{\frac{1}{2}}\eta^{\frac{1}{4}}$ and applying Lemma 2, completes the proof. $\qquad\square$

## B.2  Proof of Theorem 2

Using the decomposition (56) we have:

$$\mathbb{E}_{\epsilon,\mathcal{F}}\left[\|h\|_{L^2(\Omega;\mathbb{R})}^2\right] = \mathbb{E}_{\mathcal{F}}\left[\left\|f \circ u_{\text{enc}} - \mathbf{S}_{\lambda_{\text{d}},|\mathcal{F}|,2}[\mathbf{f}]\right\|_{L^2(\Omega;\mathbb{R})}^2\right]$$
$$+ \mathbb{E}_{\epsilon,\mathcal{F}}\left[\left\|f \circ u_{\text{enc}} - \mathbf{S}_{\lambda_{\text{d}},|\mathcal{F}|,2}[\boldsymbol{\epsilon}]\right\|_{L^2(\Omega;\mathbb{R})}^2\right],$$
(89)

where $\mathbf{f} = \left\{f(u_{\text{enc}}(\beta_{i_1})),\ldots,f(u_{\text{enc}}(\beta_{i_{|\mathcal{F}|}}))\right\}$ and $\boldsymbol{\epsilon} = \left\{\epsilon_{i_1},\ldots,\epsilon_{i_{|\mathcal{F}|}}\right\}$. Same as (79) we define the following variables:

$$\Delta_{\max}^{\mathcal{F}} := \max_{f\in\{0,\ldots,|\mathcal{F}|\}}\left\{\beta_{i_{f+1}} - \beta_{i_f}\right\}, \quad \Delta_{\min}^{\mathcal{F}} := \min_{f\in\{1,\ldots,|\mathcal{F}|-1\}}\left\{\beta_{i_{f+1}} - \beta_{i_f}\right\}.$$
(90)

Again we have $\Delta_{\max}^{\mathcal{F}} \leqslant (S+1)\cdot\Delta_{\max}$ and $\frac{\Delta_{\max}^{\mathcal{F}}}{\Delta_{\min}^{\mathcal{F}}} \leqslant (S+1)\cdot\frac{\Delta_{\max}}{\Delta_{\min}} \overset{(a)}{\leqslant} (S+1)B$, where (a) is because of the bounded condition that we have. Therefore, $\frac{\Delta_{\max}^{\mathcal{F}}}{\Delta_{\min}^{\mathcal{F}}}$ is bounded. Additionally, since $\Delta_{\min} \leqslant \frac{2}{N}$,

then $\frac{\Delta_{\max}}{\Delta_{\min}} \leqslant B$ implies that both $\Delta_{\max} = \mathcal{O}(\frac{1}{N})$. Thus, there exists a constant $J$ such that $\Delta_{\max} \leqslant \frac{J}{N}$. Therefore, we have $\Delta_{\max}^{\mathcal{F}} \leqslant \Delta_{\max} \cdot (S+1) \leqslant \frac{J(S+1)}{N} \leqslant \frac{J(S+1)}{N-S}$. Applying Theorems 6 and 7 with $\Omega = (-1, 1), m = 2$, we have:

$$\mathbb{E}_{\epsilon, \mathcal{F}}\left[\|h\|_{L^2(\Omega;\mathbb{R})}^2\right] \leqslant H_0 \left\|(f \circ u_{\text{enc}})^{(2)}\right\|_{L^2(\Omega;\mathbb{R})}^2 \cdot \mathbb{E}_{\mathcal{F}}[L] + \frac{Q_0 \sigma_0^2}{N-S} \lambda_{\text{d}}^{-\frac{1}{4}}, \tag{91}$$

and

$$\mathbb{E}_{\epsilon, \mathcal{F}}\left[\|h'\|_{L^2(\Omega;\mathbb{R})}^2\right] \leqslant H_1 \left\|(f \circ u_{\text{enc}})^{(2)}\right\|_{L^2(\Omega;\mathbb{R})}^2 \cdot \mathbb{E}_{\mathcal{F}}\left[L^{\frac{1}{2}}(1 + \frac{L}{16})^{\frac{1}{2}}\right] + \frac{Q_1 \sigma_0^2}{N-S} \lambda_{\text{d}}^{-\frac{3}{4}}, \tag{92}$$

where $L = p_2\left(\frac{\Delta_{\max}^{\mathcal{F}}}{\Delta_{\min}^{\mathcal{F}}}\right) \cdot \frac{(N-S)\Delta_{\max}^{\mathcal{F}}}{4} \lambda_{\text{d}} + D(2) \cdot \left(\Delta_{\max}^{\mathcal{F}}\right)^4$, $H_0, H_1 := H(2, 0), H(2, 1)$, and $Q_0(\lambda_0), Q_1(\lambda_0) := Q(2, 0, \lambda_0), Q(2, 1, \lambda_0)$ as defined in Theorems 6 and 7. Since $\frac{\Delta_{\max}^{\mathcal{F}}}{\Delta_{\min}^{\mathcal{F}}} \leqslant (S+1)B$, we have:

$$\begin{aligned}
\mathbb{E}_{\mathcal{F}}[L] &\leqslant \mathbb{E}_{\mathcal{F}}\left[\tilde{p_2}(S+1) \cdot \frac{(N-S)\Delta_{\max}^{\mathcal{F}}}{4} \lambda_{\text{d}} + D \cdot \left(\Delta_{\max}^{\mathcal{F}}\right)^4\right] \\
&\stackrel{\text{(a)}}{\leqslant} \tilde{p_2}(S+1) \cdot \frac{J(S+1)}{4} \lambda_{\text{d}} + DJ^4 \frac{(S+1)^4}{(N-S)^4} \\
&= p_3(S+1) \cdot \lambda_{\text{d}} + DJ^4 \frac{(S+1)^4}{(N-S)^4}, \tag{93}
\end{aligned}$$

where $D := D(2), \tilde{p_2}(S+1) = p_2(B(S+1)), p_3(S+1) := \tilde{p_2}(S+1) \cdot \frac{J(S+1)}{4}$ is a degree three polynomial of $(S+1)$, and (a) follows from the $\Delta_{\max}^{\mathcal{F}} \leqslant \frac{J(S+1)}{N-S}$. Substituting in (91), we have:

$$\begin{aligned}
\mathbb{E}_{\epsilon, \mathcal{F}}\left[\|h\|_{L^2(\Omega;\mathbb{R})}^2\right] &\leqslant H_0 \left\|(f \circ u_{\text{enc}})^{(2)}\right\|_{L^2(\Omega;\mathbb{R})}^2 \left(p_3(S+1) \cdot \lambda_{\text{d}} + DJ^4 \frac{(S+1)^4}{(N-S)^4}\right) + \frac{Q_0(\lambda_0)\sigma_0^2}{N-S} \lambda_{\text{d}}^{-\frac{1}{4}} \\
&\stackrel{\text{(a)}}{\leqslant} H_0 \left\|(f \circ u_{\text{enc}})^{(2)}\right\|_{L^2(\Omega;\mathbb{R})}^2 \lambda_{\text{d}} \cdot \left(p_3(S+1) + DJ^4(S+1)^4\right) + \frac{Q_0(\lambda_0)\sigma_0^2}{N-S} \lambda_{\text{d}}^{-\frac{1}{4}} \\
&\stackrel{\text{(b)}}{=} H_0 \left\|(f \circ u_{\text{enc}})^{(2)}\right\|_{L^2(\Omega;\mathbb{R})}^2 \lambda_{\text{d}} \cdot p_4(S+1) + \frac{Q_0(\lambda_0)\sigma_0^2}{N-S} \lambda_{\text{d}}^{-\frac{1}{4}}, \tag{94}
\end{aligned}$$

where (a) follows from the fact that $\lambda_{\text{d}}^{-1}(N-S)^{-4} \leqslant 1$, as assumed in Theorem 7 and (b) is by definition $p_4(S+1) := p_3(S+1) + DJ^4(S+1)^4$ is a degree four polynomial of $(S+1)$. An analogous upper bound can be derived for $\mathbb{E}_{\epsilon, \mathcal{F}}\left[\|h'\|_{L^2(\Omega;\mathbb{R})}^2\right]$ as

$$\begin{aligned}
\mathbb{E}_{\epsilon, \mathcal{F}}\left[\|h'\|_{L^2(\Omega;\mathbb{R})}^2\right] &\leqslant H_1 \left\|(f \circ u_{\text{enc}})^{(2)}\right\|_{L^2(\Omega;\mathbb{R})}^2 \cdot \lambda_{\text{d}}^{\frac{1}{2}} \cdot p_4(S+1)^{\frac{1}{2}} \cdot \left(1 + \lambda_{\text{d}} \frac{p_4(S+1)}{16}\right)^{\frac{1}{2}} \\
&\qquad\qquad + \frac{Q_1(\lambda_0)\sigma_0^2}{N-S} \lambda_{\text{d}}^{-\frac{3}{4}} \\
&\stackrel{\text{(a)}}{\leqslant} H_1 \left\|(f \circ u_{\text{enc}})^{(2)}\right\|_{L^2(\Omega;\mathbb{R})}^2 \cdot \lambda_{\text{d}}^{\frac{1}{2}} \cdot p_4(S+1)^{\frac{1}{2}} \cdot \left(1 + \lambda_0 \frac{p_4(S+1)}{16}\right)^{\frac{1}{2}} \\
&\qquad\qquad + \frac{Q_1(\lambda_0)\sigma_0^2}{N-S} \lambda_{\text{d}}^{-\frac{3}{4}} \\
&\stackrel{\text{(b)}}{=} \left\|(f \circ u_{\text{enc}})^{(2)}\right\|_{L^2(\Omega;\mathbb{R})}^2 \tilde{\eta} \cdot \lambda_{\text{d}}^{\frac{1}{2}} \cdot p_4(S+1)^{\frac{1}{2}} + \frac{Q_1(\lambda_0)\sigma_0^2}{N-S} \lambda_{\text{d}}^{-\frac{3}{4}}, \tag{95}
\end{aligned}$$

where (a) follows from the definition $\lambda_{\text{d}} \leqslant \lambda_0$, (b) is derived from the definition $\tilde{\eta} := H_1 \left(1 + \lambda_0 \frac{p_4(S+1)}{16}\right)^{\frac{1}{2}}$. By applying the upper bound for $\mathcal{L}_{\text{dec}}$ from (77) and incorporating the

results from (94) and (95), we can deduce the following:

$$
\mathcal{L}_{\text{dec}} \overset{(a)}{\leqslant} 2 \left( \left\| (f \circ u_{\text{enc}})^{(2)} \right\|^2_{L^2(\Omega;\mathbb{R})} \cdot \mu_0(S)\lambda_{\text{d}} + \frac{Q_0(\lambda_0)\sigma_0^2}{N-S} \lambda_{\text{d}}^{-\frac{1}{4}} \right)^{\frac{1}{2}}
$$

$$
\cdot \left( \left\| (f \circ u_{\text{enc}})^{(2)} \right\|^2_{L^2(\Omega;\mathbb{R})} \cdot \mu_1(S)\lambda_{\text{d}}^{\frac{1}{2}} + \frac{Q_1(\lambda_0)\sigma_0^2}{N-S} \lambda_{\text{d}}^{-\frac{3}{4}} \right)^{\frac{1}{2}}
$$

$$
\overset{(b)}{\leqslant} 2\lambda_{\text{d}}^{\frac{1}{4}} \left( \left\| (f \circ u_{\text{enc}})^{(2)} \right\|^2_{L^2(\Omega;\mathbb{R})} \cdot \mu_{\max}(S)\lambda_{\text{d}}^{\frac{1}{2}} + \frac{Q_{\max}(\lambda_0)\sigma_0^2}{N-S} \lambda_{\text{d}}^{-\frac{3}{4}} \right)
$$

$$
= 2\lambda_{\text{d}}^{\frac{3}{4}} \cdot \left\| (f \circ u_{\text{enc}})^{(2)} \right\|^2_{L^2(\Omega;\mathbb{R})} \cdot \mu_{\max}(S) + \frac{Q_{\max}(\lambda_0)\sigma_0^2}{N-S} \lambda_{\text{d}}^{-\frac{1}{2}}, \tag{96}
$$

where (a) follows by the definitions $\mu_0(S) := H_0 \cdot p_4(S+1)$ and $\mu_1(S) := \tilde{\eta} \cdot p_4(S+1)^{\frac{1}{2}}$, and (b) is due to $Q_0(\lambda_0), Q_1(\lambda_0) \leqslant Q_{\max}(\lambda_0) := \max\{Q_0(\lambda_0), Q_1(\lambda_0)\}$ and $\mu_0(S), \mu_1(S) \leqslant \mu_{\max}(S) := \max\{\mu_0(S), \mu_1(S)\}$. Therefore, we can conclude that:

$$
\mathcal{L}_{\text{dec}} \leqslant 2\lambda_{\text{d}}^{\frac{3}{4}} \cdot \mu_{\max}(S) \cdot \left\| (f \circ u_{\text{enc}})^{(2)} \right\|^2_{L^2(\Omega;\mathbb{R})} + \lambda_{\text{d}}^{-\frac{1}{2}} \cdot \frac{Q_{\max}(\lambda_0)\sigma_0^2}{N-S}. \tag{97}
$$

Based on the definition of $\mu_{\max}(S)$ we have:

$$
\mu_{\max}(S) = \max \left\{ H_0 \cdot p_4(S), H_1 \left( 1 + \lambda_0 \frac{p_4(S)}{16} \right)^{\frac{1}{2}} p_4(S)^{\frac{1}{2}} \right\}
$$

$$
\leqslant H_{\max} \cdot p_4(S)^{\frac{1}{2}} \cdot \max \left\{ p_4(S), 1 + \frac{\lambda_0 p_4(S)}{16} \right\}^{\frac{1}{2}},
$$

$$
\leqslant H_{\max} \cdot p_4(S)^{\frac{1}{2}} \cdot \left( \frac{1 + p_4(S)}{16} \right)^{\frac{1}{2}} \max\{16, \lambda_0\}^{\frac{1}{2}},
$$

$$
= H_{\max} \cdot \widetilde{p_4}(S) \max\{4, \lambda_0\}^{\frac{1}{2}}, \tag{98}
$$

where $H_{\max} := \max\{H_0, H_1\}$ and $\widetilde{p_4}(S) := p_4(S)^{\frac{1}{2}} \cdot \left( \frac{1+p_4(S)}{16} \right)^{\frac{1}{2}}$. Based on definition of $Q_{\max}(\lambda_0)$ (mentioned in Theorem 7), we have:

$$
Q_{\max}(\lambda_0) = \max\{w_0\lambda_0^{\frac{1}{4}} + \tilde{w}_0, w_1\lambda_0^{\frac{1}{4}} + \tilde{w}_1\}
$$

$$
\leqslant w_{\max}\lambda_0^{\frac{1}{4}} + \tilde{w}_{\max},
$$

$$
\leqslant 2w_{\max} \max \left\{ \lambda_0, \left( \frac{\tilde{w}_{\max}}{w_{\max}} \right)^4 \right\}^{\frac{1}{4}} \tag{99}
$$

where $w_{\max} := \max\{w_0, w_1\}$ and $\tilde{w}_{\max} := \max\{\tilde{w}_0, \tilde{w}_1\}$. Therefore we have:

$$
\mathcal{L}_{\text{dec}} \leqslant 2\lambda_{\text{d}}^{\frac{3}{4}} \cdot H_{\max} \cdot \widetilde{p_4}(S) \max\{4, \lambda_0\}^{\frac{1}{2}} \cdot \left\| (f \circ u_{\text{enc}})^{(2)} \right\|^2_{L^2(\Omega;\mathbb{R})} + \lambda_{\text{d}}^{-\frac{1}{2}} \frac{2\sigma_0^2 w_{\max} \max\left\{ \lambda_0^{\frac{1}{4}}, \frac{\tilde{w}_{\max}}{w_{\max}} \right\}}{N-S}. \tag{100}
$$

Since (97) holds for all $\lambda_{\text{d}}$, by minimizing the right-hand side of (97) with respect to $\lambda_{\text{d}}$, we obtain:

$$
\lambda_{\text{d}}^* = \left( \frac{3H_{\max} \cdot \widetilde{p_4}(S) \cdot (N-S) \cdot \left\| (f \circ u_{\text{enc}})^{(2)} \right\|^2_{L^2(\Omega;\mathbb{R})}}{4w_{\max}\sigma_0^2} \right)^{-\frac{4}{5}} \left( \frac{\max\{4, \lambda_0\}^{\frac{1}{2}}}{\max\left\{ \lambda_0^{\frac{1}{4}}, \frac{\tilde{w}_{\max}}{w_{\max}} \right\}} \right)^{-\frac{4}{5}} \tag{101}
$$

By substituting the expression for $\lambda_{\text{d}}^*$ into (97), we have:

$$
\mathcal{L}_{\text{dec}} \leqslant 4 \left( \frac{3H_{\max}}{4w_{\max}} \right)^{\frac{2}{5}} \left( \frac{\sigma_0}{N-S} \right)^{\frac{3}{5}} \cdot \widetilde{p_4}(S)^{\frac{2}{5}} \cdot \left\| (f \circ u_{\text{enc}})^{(2)} \right\|^{\frac{4}{5}}_{L^2(\Omega;\mathbb{R})} \left( \frac{\max\{4, \lambda_0\}^{\frac{1}{2}}}{\max\left\{ \lambda_0^{\frac{1}{4}}, \frac{\tilde{w}_{\max}}{w_{\max}} \right\}} \right)^{\frac{2}{5}}. \tag{102}
$$

Thus, defining $C_2 := \frac{3H_{\max}}{4w_{\max}}, C(\lambda_0) := \left( \frac{\max\{4,\lambda_0\}^{\frac{1}{2}}}{\max\left\{\lambda_0^{\frac{1}{4}}, \frac{\tilde{w}_{\max}}{w_{\max}}\right\}} \right)$, and using previously driven upper bound

for $\mathcal{L}_{\text{enc}}$ in Lemma 2 completes the proof.

## B.3 Proof of Theorem 3

The upper bounds presented in Theorems 1 and 2 depend on $\left\|(f \circ u_{\text{enc}})^{(2)}\right\|_{L^2(\Omega;\mathbb{R})}^2$, with exponents of $\frac{2}{5}$ and 1, respectively. By applying the chain rule, we can demonstrate that:

$$
\begin{aligned}
\int_{\Omega} \left[f(u_{\text{enc}}(t))''\right]^2 dt &\overset{(a)}{=} \int_{\Omega} \left[u_{\text{enc}}''(t) \cdot f'(u_{\text{enc}}(t)) + u_{\text{enc}}'(t)^2 \cdot f''(u_{\text{enc}}(t))\right]^2 dt \\
&\overset{(b)}{\leqslant} \int_{\Omega} \left[u_{\text{enc}}''(t)^2 + u_{\text{enc}}'(t)^4\right]\left[f'(u_{\text{enc}}(t))^2 + f''(u_{\text{enc}}(t))^2\right] dt \\
&\overset{(c)}{\leqslant} (q^2 + \nu^2) \int_{\Omega} \left[u_{\text{enc}}''(t)^2 + u_{\text{enc}}'(t)^4\right] dt \\
&= (q^2 + \nu^2) \left(\|u_{\text{enc}}''(t)\|_{L^2(\Omega;\mathbb{R})}^2 + \|u_{\text{enc}}'(t)\|_{L^4(\Omega;\mathbb{R})}^4\right) \\
&\overset{(d)}{\leqslant} (q^2 + \nu^2) \left(\|u_{\text{enc}}''(t)\|_{L^2(\Omega;\mathbb{R})}^2 + \left(\|u_{\text{enc}}'(t)\|_{L^2(\Omega;\mathbb{R})} + \|u_{\text{enc}}''(t)\|_{L^2(\Omega;\mathbb{R})}\right)^4\right) \\
&\overset{(e)}{\leqslant} (q^2 + \nu^2) \left(\|u_{\text{enc}}''(t)\|_{L^2(\Omega;\mathbb{R})}^2 + 4\left(\|u_{\text{enc}}'(t)\|_{L^2(\Omega;\mathbb{R})}^2 + \|u_{\text{enc}}''(t)\|_{L^2(\Omega;\mathbb{R})}^2\right)^2\right) \\
&\overset{(f)}{\leqslant} (q^2 + \nu^2) \left(\|u_{\text{enc}}\|_{\mathbb{W}^{2,2}(\Omega;\mathbb{R})}^2 + 4\|u_{\text{enc}}\|_{\mathbb{W}^{2,2}(\Omega;\mathbb{R})}^4\right) \\
&\overset{(g)}{\leqslant} (q^2 + \nu^2) \left(73\|u_{\text{enc}}\|_{\widetilde{\mathbb{W}}^{2,2}(\Omega;\mathbb{R})}^2 + 4 \times 73^2 \|u_{\text{enc}}\|_{\widetilde{\mathbb{W}}^{2,2}(\Omega;\mathbb{R})}^4\right) \\
&\overset{(h)}{=} (q^2 + \nu^2) \cdot \psi\left(\|u_{\text{enc}}\|_{\widetilde{\mathbb{W}}^{2,2}(\Omega;\mathbb{R})}^2\right), \\
&\overset{(i)}{=} (q^2 + \nu^2) \cdot \psi\left(\|u_{\text{enc}}\|_{\widetilde{\mathcal{H}}^2(\Omega;\mathbb{R})}^2\right),
\end{aligned}
\tag{103}
$$

where (a) follows from the chain rule, (b) is derived using the Cauchy-Schwartz inequality, (c) is due to $\|f_0''\|_{L^\infty(\Omega;\mathbb{R})} \leqslant \nu$, (d) follows from Theorem 5 with $r = 4, p = q = 2, l = 1$, (e) follows from AM-GM inequality, (f) is a result of adding positive terms $\|u_{\text{enc}}\|_{\mathbb{W}^{2,2}(\Omega;\mathbb{R})}^2$ and $\|u_{\text{enc}}'\|_{\mathbb{W}^{2,2}(\Omega;\mathbb{R})}^2$ to the first term and $\|u_{\text{enc}}\|_{\mathbb{W}^{2,2}(\Omega;\mathbb{R})}^2$ to the second term in the parenthesis, (g) is result of applying Corollary 4, (h) is by defining $\psi(t) := 73t + 4 \times 73^2 t^2$, and (i) is because of Proposition 2.

Combining (103) with Theorems 1 and 2 we have

$$
\mathcal{R}(\hat{f}) \leqslant \frac{2q^2}{K} \sum_{k=1}^{K} (u_{\text{enc}}(\alpha_k) - x_k)^2 + \lambda_e \cdot \psi\left(\|u_{\text{enc}}\|_{\widetilde{\mathcal{H}}^2(\Omega;\mathbb{R})}^2\right),
\tag{104}
$$

for the noiseless setting and

$$
\mathcal{R}(\hat{f}) \leqslant \frac{2q^2}{K} \sum_{k=1}^{K} (u_{\text{enc}}(\alpha_k) - x_k)^2 + \widetilde{\lambda}_e \cdot \psi\left(\|u_{\text{enc}}\|_{\widetilde{\mathcal{H}}^2(\Omega;\mathbb{R})}^2\right)^{\frac{2}{5}}
\tag{105}
$$

for the noisy setting, where the parameters $\lambda_e$ and $\widetilde{\lambda}_e$ are as follows:

$$
\lambda_e = C_1 \frac{(S+1)^3}{N^3} \cdot (q^2 + \nu^2)
\tag{106}
$$

$$
\widetilde{\lambda}_e = 2C(\lambda_0)^{\frac{3}{4}} \left(\frac{\sigma_0^2}{N-S}\right)^{\frac{3}{5}} \cdot p_4(S)^{\frac{2}{5}} \cdot (q^2 + \nu^2)^{\frac{2}{5}}.
\tag{107}
$$

Note that since $\psi(t)$ and $\gamma(t) := t^{\frac{2}{5}}$ are monotonically increasing in $\mathbb{R}^+$, its composition is monotonically increasing as well. Moreover, $\lambda_e$ and $\widetilde{\lambda}_e$ share the same exponent of $N$ as in Theorems 1 and 2, respectively. Consequently, the provided upper bound does not compromise the convergence rate.

## B.4 Proof of Proposition 1

For part (i), we know that for $t \leqslant M$, we have:

$$\psi(t) = 73t + 4 \times 73^2 t^2 \leqslant t(73 + 4 \times 73^2 t) \leqslant t(73 + 4 \times 73^2 \cdot M) = t(m_1 + m_2 M), \quad (108)$$

where $m_1 := 73, m_2 := 4 \times 73^2$. Therefore, if $\|u_{\text{enc}}\|^2_{\widetilde{\mathcal{H}}^2(\Omega;\mathbb{R})} \leqslant M$, then:

$$\begin{aligned} \psi(\|u_{\text{enc}}\|^2_{\widetilde{\mathcal{H}}^2(\Omega;\mathbb{R})}) &\leqslant (m_1 + m_2 M) \|u_{\text{enc}}\|^2_{\widetilde{\mathcal{H}}^2(\Omega;\mathbb{R})} \\ &= (m_1 + m_2 M) \left( u_{\text{enc}}(-1)^2 + u'_{\text{enc}}(-1)^2 + \int_\Omega |u''_{\text{enc}}(t)|^2 \, dt \right) \\ &\overset{(a)}{\leqslant} (m_1 + m_2 M) \left( M + \int_\Omega |u''_{\text{enc}}(t)|^2 \, dt \right), \end{aligned} \quad (109)$$

where (a) is because of $\|u_{\text{enc}}\|^2_{\widetilde{\mathcal{H}}^2(\Omega;\mathbb{R})} \leqslant M$. Thus, we have:

$$\begin{aligned} \mathcal{R}(\hat{f}) &\leqslant \frac{2q^2}{K} \sum_{k=1}^{K} (u_{\text{enc}}(\alpha_k) - x_k)^2 + \lambda_{\text{e}} \cdot \psi \left( \|u_{\text{enc}}\|^2_{\widetilde{\mathcal{H}}^2(\Omega;\mathbb{R})} \right) \\ &\leqslant \frac{2q^2}{K} \sum_{k=1}^{K} (u_{\text{enc}}(\alpha_k) - x_k)^2 + \lambda_{\text{e}} \cdot (m_1 + m_2 M) \left( M + \int_\Omega |u''_{\text{enc}}(t)|^2 \, dt \right) \\ &\leqslant \widetilde{R}(u_{\text{enc}}). \end{aligned} \quad (110)$$

For part (ii), let $\widetilde{u_{\text{enc}}}(t)$ be a natural spline used as the encoder function fitted to the data points $\{(\alpha_k, x_k)\}_{k=1}^{K}$. Then, we have:

$$\begin{aligned} \lambda_{\text{e}}(m_1 + m_2 M) \left( M + \int_\Omega |(u^*)''(t)|^2 \, dt \right) &\leqslant \widetilde{R}(u^*) \\ &\overset{(a)}{\leqslant} \widetilde{R}(u_{\text{enc}}) \overset{(b)}{=} \lambda_{\text{e}}(m_1 + m_2 M) \left( M + \int_\Omega |\widetilde{u_{\text{enc}}}''(t)|^2 \, dt \right), \end{aligned} \quad (111)$$

where (a) follows from the optimality of $u^*(\cdot)$, and (b) follows from $\widetilde{u_{\text{enc}}}(\alpha_k) = x_k$ for $k \in [K]$. Therefore, $\int_\Omega |(u^*)''(t)|^2 \, dt \leqslant \int_\Omega |\widetilde{u_{\text{enc}}}''(t)|^2 \, dt$.

Since $u^*(\cdot)$ is smoothing spline, it has the representation in natural spline space (as mentioned in (51)):

$$u^*(t) = \sum_{k=1}^{K+4} \xi_k b_k(t), \quad (112)$$

where, $\{b_k(\cdot)\}_{k=1}^{K}$ is a basis functions of second order natural splines. Therefore, using Cauchy-Schwartz inequality, we have:

$$|u^*(t)|^2 \leqslant \left( \sum_{k=1}^{K+4} \xi^2 \right) \left( \sum_{k=1}^{K+4} |b_k(t)|^2 \right), \quad (113)$$

and

$$|(u^*(t))'|^2 \leqslant \left( \sum_{k=1}^{K+4} \xi^2 \right) \left( \sum_{k=1}^{K+4} |b'_k(t)|^2 \right). \quad (114)$$

Both (113) and (114) hold for all $t \in \Omega$. Thus:

$$|(u^*)'(-1)|^2 \leqslant \|u^*\|^2_{L^\infty(\Omega;\mathbb{R})} \leqslant \|\boldsymbol{\xi}\|_2^2 \cdot \left( \sum_{k=1}^{K+4} \|b_k\|^2_{L^\infty(\Omega;\mathbb{R})} \right), \quad (115)$$

$$|(u^*)'(-1)|^2 \leqslant \|(u^*)'\|^2_{L^\infty(\Omega;\mathbb{R})} \leqslant \|\boldsymbol{\xi}\|_2^2 \cdot \left( \sum_{k=1}^{K+4} \|b'_k\|^2_{L^\infty(\Omega;\mathbb{R})} \right), \quad (116)$$

where $\boldsymbol{\xi} := [\xi_1, \ldots, \xi_{K+4}]^T = \left(\mathbf{N}^T\mathbf{N} + \lambda\Phi\right)^{-1}\mathbf{N}^T\mathbf{x}$ with $\mathbf{N}_{ij} = b_i(\alpha_j)$, $\Phi_{ij} = \int_\Omega b_i''(t)b_j''(t)\,dt$ for $i, j \in [K+4]$ as defined in (51), and $\mathbf{x} := [x_1, \ldots, x_K]$. Noted that $\{\|b_k'\|_{L^\infty(\Omega;\mathbb{R})}^2\}_{k=1}^{K+4}$ and $\{\|b_k\|_{L^\infty(\Omega;\mathbb{R})}^2\}_{k=1}^{K+4}$ depend only on $\{\alpha_k\}_{k=1}^K$.

**Lemma 6.** *If* $\widetilde{\boldsymbol{\xi}} := \left(\mathbf{N}^T\mathbf{N}\right)^{-1}\mathbf{N}^T\mathbf{x}$, *then* $\|\boldsymbol{\xi}\|_2^2 \leqslant \kappa(\Phi) \cdot \left\|\widetilde{\boldsymbol{\xi}}\right\|_2^2 < \frac{2}{|\det\Phi|}\left(\frac{\|\Phi\|_F^2}{K}\right)^{\frac{K}{2}}\left\|\widetilde{\boldsymbol{\xi}}\right\|_2^2$, *where* $\kappa(\Phi)$ *is condition number of* $\Phi$.

*Proof.* By defining $\widetilde{\mathbf{N}} := \mathbf{N}\Phi^{-\frac{1}{2}}$ and rearranging the expression for $\widetilde{\boldsymbol{\xi}}$, we obtain:

$$\widetilde{\boldsymbol{\xi}} = \left(\mathbf{N}^T\mathbf{N} + \lambda\Phi\right)^{-1}\mathbf{N}^T\mathbf{x} = \Phi^{-1/2}\left(\left[\mathbf{N}\Phi^{-1/2}\right]^T\left[\mathbf{N}\Phi^{-1/2}\right] + \lambda\mathbf{I}\right)^{-1}\left[\mathbf{N}\Phi^{-1/2}\right]^T\mathbf{x}$$

$$= \Phi^{-1/2}\left(\widetilde{\mathbf{N}}^T\widetilde{\mathbf{N}} + \lambda\mathbf{I}\right)^{-1}\widetilde{\mathbf{N}}^T\mathbf{x}. \tag{117}$$

Define $\mathbf{z} := \left(\widetilde{\mathbf{N}}^T\widetilde{\mathbf{N}}\right)^{-1}\widetilde{\mathbf{N}}^T\mathbf{x}$. Thus, $\widetilde{\mathbf{N}}^T\mathbf{x} = \widetilde{\mathbf{N}}^T\widetilde{\mathbf{N}}\mathbf{z}$. Thus, by applying the Cauchy-Schwartz inequality, we have:

$$\left\|\left(\widetilde{\mathbf{N}}^T\widetilde{\mathbf{N}} + \lambda\mathbf{I}\right)^{-1}\widetilde{\mathbf{N}}^T\mathbf{x}\right\|_2 = \left\|\left(\widetilde{\mathbf{N}}^T\widetilde{\mathbf{N}} + \lambda\mathbf{I}\right)^{-1}\widetilde{\mathbf{N}}^T\widetilde{\mathbf{N}}\mathbf{z}\right\|_2 \leqslant \left\|\left(\widetilde{\mathbf{N}}^T\widetilde{\mathbf{N}} + \lambda\mathbf{I}\right)^{-1}\widetilde{\mathbf{N}}^T\widetilde{\mathbf{N}}\right\|_2 \cdot \|\mathbf{z}\|_2. \tag{118}$$

Let $\widetilde{\mathbf{N}} = \mathbf{U}\mathbf{D}\mathbf{V}^T$ be the singular value decomposition of $\widetilde{\mathbf{N}}$. Therefore, we have:

$$\left\|\left(\widetilde{\mathbf{N}}^T\widetilde{\mathbf{N}} + \lambda\mathbf{I}\right)^{-1}\widetilde{\mathbf{N}}^T\widetilde{\mathbf{N}}\right\|_2 = \left\|\left(\mathbf{V}\mathbf{D}^2\mathbf{V}^T + \lambda\mathbf{I}\right)^{-1}\mathbf{V}\mathbf{D}^2\mathbf{V}^T\right\|_2$$

$$= \left\|\mathbf{V}^{-T}\left(\mathbf{D}^2 + \lambda\mathbf{I}\right)^{-1}\mathbf{V}^{-1}\mathbf{V}\mathbf{D}^2\mathbf{V}^T\right\|_2$$

$$= \left\|\mathbf{V}^{-T}\left(\mathbf{D}^2 + \lambda\mathbf{I}\right)^{-1}\mathbf{D}^2\mathbf{V}^T\right\|_2$$

$$\overset{(a)}{=} \left\|\left(\mathbf{D}^2 + \lambda\mathbf{I}\right)^{-1}\mathbf{D}^2\right\|_2$$

$$= \left\|\text{diag}\left(\frac{\lambda_1^2}{\lambda_1^2 + \lambda}, \ldots, \frac{\lambda_K^2}{\lambda_K^2 + \lambda}\right)\right\|_2$$

$$\leqslant 1, \tag{119}$$

where (a) is because $\mathbf{V}$ is an unitary matrix and $\lambda_1, \ldots, \lambda_K$ are eigenvalues of $\widetilde{\mathbf{N}}$. Continuing from (117), we obtain:

$$\left\|\Phi^{1/2}\widetilde{\boldsymbol{\xi}}\right\|_2 = \left\|\left(\widetilde{\mathbf{N}}^T\widetilde{\mathbf{N}} + \lambda\mathbf{I}\right)^{-1}\widetilde{\mathbf{N}}^T\widetilde{\mathbf{N}}\right\|_2 \leqslant \|\mathbf{z}\|_2. \tag{120}$$

Let us define $\mathbf{x}_0 := \left(\mathbf{N}^T\mathbf{N}\right)^{-1}\mathbf{N}^T\mathbf{x}$. Thus, we have:

$$\mathbf{z} = \left(\widetilde{\mathbf{N}}^T\widetilde{\mathbf{N}}\right)^{-1}\widetilde{\mathbf{N}}^T\mathbf{x}$$

$$= \left(\Phi^{-1/2}\mathbf{N}^T\mathbf{N}\Phi^{-1/2}\right)^{-1}\Phi^{-1/2}\mathbf{N}^T\mathbf{x}$$

$$= \Phi^{1/2}\left(\mathbf{N}^T\mathbf{N}\right)^{-1}\mathbf{N}^T\mathbf{x}$$

$$= \Phi^{1/2}\mathbf{x}_0. \tag{121}$$

Therefore, we can bound the $\|\boldsymbol{\xi}\|_\Phi := \sqrt{\boldsymbol{\xi}^T\Phi\boldsymbol{\xi}}$:

$$\|\boldsymbol{\xi}\|_\Phi = \left\|\Phi^{1/2}\boldsymbol{\xi}\right\|_2 \leqslant \|\mathbf{z}\|_2 = \|\mathbf{x}_0\|_\Phi. \tag{122}$$

Since $\Phi$ is symmetric, by Rayleigh-Ritz theorem we know that:

$$0 \overset{(a)}{<} \lambda_{\min}^{\Phi} \leqslant \frac{\boldsymbol{\xi}^T \Phi \boldsymbol{\xi}}{\boldsymbol{\xi}^T \boldsymbol{\xi}} = \frac{\|\boldsymbol{\xi}\|_{\Phi}^2}{\|\boldsymbol{\xi}\|_2^2} \leqslant \lambda_{\max}^{\Phi}, \tag{123}$$

where $\lambda_{\min}^{\Phi}, \lambda_{\max}^{\Phi}$ are minimum and maximum eigenvalues of $\Phi$, and (a) is due to the fact that since $\Phi$ is kernel matrix of RKHS space and $\{b_k(\cdot)\}_{k=1}^{K}$ are basis functions, it is positive definite. Thus, we have:

$$\|\boldsymbol{\xi}\|_2^2 \leqslant \frac{1}{\lambda_{\min}^{\Phi}} \|\boldsymbol{\xi}\|_{\Phi}^2 \leqslant \frac{1}{\lambda_{\min}^{\Phi}} \cdot \|\mathbf{x}_0\|_{\Phi}^2 \leqslant \frac{\lambda_{\max}^{\Phi}}{\lambda_{\min}^{\Phi}} \|\mathbf{x}_0\|_2^2 = \kappa(\Phi) \|\mathbf{x}_0\|_2^2. \tag{124}$$

Applying the bound for condition number introduce in [66], we can complete the proof:

$$\kappa(\Phi) < \frac{2}{|\det \Phi|} \left( \frac{\|\Phi\|_F^2}{K+4} \right)^{\frac{K+4}{2}}, \tag{125}$$

where $\|\cdot\|_F$ is the Frobenius norm. $\square$

Using Lemma 6, (115), and (116) we have:

$$
\begin{aligned}
\|u^*\|_{\widetilde{\mathcal{H}}^2(\Omega;\mathbb{R})} &\leqslant \|\boldsymbol{\xi}\|_2^2 \cdot \left( \sum_{k=1}^{K+4} \|b_k\|_{L^\infty(\Omega;\mathbb{R})}^2 + \sum_{k=1}^{K+4} \|b_k'\|_{L^\infty(\Omega;\mathbb{R})}^2 \right) + \int_\Omega |\widetilde{u_{\text{enc}}}''(t)|^2 \, dt \\
&\overset{(a)}{\leqslant} \frac{2}{|\det \Phi|} \left( \frac{\|\Phi\|_F^2}{K+4} \right)^{\frac{K+4}{2}} \left\|\widetilde{\boldsymbol{\xi}}\right\|_2^2 \left( \sum_{k=1}^{K+4} \|b_k\|_{L^\infty(\Omega;\mathbb{R})}^2 + \sum_{k=1}^{K+4} \|b_k'\|_{L^\infty(\Omega;\mathbb{R})}^2 \right) \\
&\quad + \int_\Omega |\widetilde{u_{\text{enc}}}''(t)|^2,
\end{aligned}
\tag{126}
$$

where (a) follows by applying Lemma 6. Setting $M$ equal to the right-hand side of Equation (126) completes the proof.

### B.5 Proof of Theorem 4

Consider a natural spline $\widetilde{u_{\text{enc}}}(t)$ as the encoder function fitted on the data points $\{(\alpha_k, x_k)\}_{k=1}^{K}$. Let $u_{\text{enc}}^*(t)$ denote the optimal encoder minimizing the upper bound in (10). Then, we have:

$$
\begin{aligned}
\mathcal{R}(\hat{f}) &\leqslant \frac{2q^2}{K} \sum_{k=1}^{K} (u_{\text{enc}}^*(\alpha_k) - x_k)^2 + \lambda_{\text{e}} \cdot g\big( \|u_{\text{enc}}^*\|_{\widetilde{\mathcal{H}}^2(\Omega;\mathbb{R})}^2 \big) \\
&\overset{(a)}{\leqslant} \frac{2q^2}{K} \sum_{k=1}^{K} (\widetilde{u_{\text{enc}}}(\alpha_k) - x_k)^2 + \lambda_{\text{e}} \cdot g\big( \|\widetilde{u_{\text{enc}}}\|_{\widetilde{\mathcal{H}}^2(\Omega;\mathbb{R})}^2 \big) \\
&\overset{(b)}{=} \lambda_{\text{e}} \cdot g\big( \|\widetilde{u_{\text{enc}}}\|_{\widetilde{\mathcal{H}}^2(\Omega;\mathbb{R})}^2 \big),
\end{aligned}
\tag{127}
$$

where (a) follows from the optimality of $u_{\text{enc}}^*(t)$, and (b) is due to the fact that $\widetilde{u_{\text{enc}}}(\alpha_k) = x_k$, since $\widetilde{u_{\text{enc}}}(\cdot)$ is a natural spline. Note that $g\big( \|\widetilde{u_{\text{enc}}}\|_{\widetilde{\mathcal{H}}^2(\Omega;\mathbb{R})}^2 \big)$ is independent of $N$ and $S$, and depends only on $\alpha_k$ and $x_k$ for $k \in [K]$. Additionally, based on the Theorem 3 and (106), $\lambda_{\text{e}} = \mathcal{O}(S^3 N^{-3})$ and $\lambda_{\text{e}} = \mathcal{O}(S^{\frac{8}{5}} N^{-\frac{3}{5}})$ for the noiseless and noisy cases, respectively. Thus, the upper bound provided in (10) converges at most at the rate of $\mathcal{O}(S^3 N^{-3})$ for the noiseless case and $\mathcal{O}(S^{\frac{8}{5}} N^{-\frac{3}{5}})$ for the noisy case.

## C Comparison with Berrut Coded Computing

### C.1 Convergence rate

The upper bound of the infinity norm for the estimation provided in [29] for the coded computing scheme with $N$ workers and maximum of $S$ stragglers is as follows:

$$\left\| \hat{f}_{\text{BACC}}(t) - f \circ u_{\text{enc}}(t) \right\|_{L^\infty(\Omega;\mathbb{R})} \leqslant 2(1+R) \sin\left( \frac{(S+1)\pi}{2N} \right) \|f \circ u_{\text{enc}}''(t)\|_{L^\infty(\Omega;\mathbb{R})}, \tag{128}$$

if $N - s$ is odd, and

$$\left\|\hat{f}_{\text{BACC}}(t) - f \circ u_{\text{enc}}(t)\right\|_{L^{\infty}(\Omega;\mathbb{R})} \leqslant 2(1 + R)\sin\left(\frac{(S+1)\pi}{2N}\right)\Big(\left\|f \circ u_{\text{enc}}''(t)\right\|_{L^{\infty}(\Omega;\mathbb{R})}$$
$$+ \left\|f \circ u_{\text{enc}}'(t)\right\|_{L^{\infty}(\Omega;\mathbb{R})}\Big), \quad (129)$$

if $N - s$ is even, where $R = \frac{(s+1)(s+3)\pi^2}{4}$. Since $\|\cdot\|_{L^2(\Omega;\mathbb{R})}$ is upper bounded by $\|\cdot\|_{L^{\infty}(\Omega;\mathbb{R})}$, we can directly derive a convergence rate for the squared $L^2(\Omega;\mathbb{R})$-norm of the error as $N$ increases:

$$\left\|\hat{f}_{\text{BACC}}(t) - f \circ u_{\text{enc}}(t)\right\|_{L^2(\Omega;\mathbb{R})}^2 \leqslant \mathcal{L}(\Omega) \cdot \left\|\hat{f}_{\text{BACC}}(t) - f \circ u_{\text{enc}}(t)\right\|_{L^{\infty}(\Omega;\mathbb{R})}^2 \leqslant \mathcal{O}(S^4 N^{-2}). \quad (130)$$

Compared to our results, the upper bound for `LeTCC` provided in Theorem 1, $\mathcal{O}(S^3 N^{-3})$, is less sensitive to the number of stragglers and converges faster with increasing $N$. Note that, since the $\|\cdot\|_{L^2(\Omega;\mathbb{R})}$ is upper bounded by $\|\cdot\|_{L^{\infty}(\Omega;\mathbb{R})}$, the statement above does not guarantee faster convergence of the proposed scheme compared to Berrut approach.

It should be noted that [29] does not analyze the noisy setting.

### C.2 Computational complexity

From the experimental view, we compare the whole encoding and decoding time (on a single CPU machine) for `LeTCC` and `BACC` frameworks, as shown in the following table:

Table 2: Average and std of end-to-end processing time of `LeTCC` and `BACC` for different architectures

|  | BACC | LeTCC |
|---|---|---|
| LeNet5, $(N, K) = (100, 20)$ | $0.013s \pm 0.002$ | $0.007s \pm 0.001$ |
| RepVGG, $(N, K) = (60, 20)$ | $1.62s \pm 0.18$ | $1.59s \pm 0.14$ |
| ViT, $(N, K) = (20, 8)$ | $1.60s \pm 0.28$ | $1.74s \pm 0.29$ |

As shown in Table 2, the end-to-end processing time of the proposed framework is on par with `BACC`.

## D Comparison with Lagrange Coded Computing

Although the **only** existing coded computing scheme for general functions is Berrut coded computing [29], with which we have compared our proposed scheme, other schemes are designed for specific computations, such as polynomial functions [3] and matrix multiplication [13]. To provide further comparison, we evaluate our proposed scheme against Lagrange coded computing (LCC) [3], which is specifically designed for polynomial computations, as follows:

### D.1 Accuracy of function approximation

LCC is applicable only to polynomial computing functions [3]. Additionally, to enable recovery, the number of servers required must be at least $(K - 1) \times \deg(f) + S + 1$ worker nodes [3, 29]; otherwise, the master node cannot recover any results. Moreover, LCC is designed for computation over finite fields and encounters serious instability when computing over real numbers, particularly when $(K - 1) \times \deg(f)$ is around 25 or higher [18, 29].

We compare the proposed framework with LCC in Figure 5. Note that if $N < (K-1) \times \deg(f) + S + 1$, LCC cannot operate effectively. To adapt LCC for such cases, we approximate results by fitting a lower-degree polynomial to the available workers' outputs. We run `LeTCC` and LCC on the same set of input data and a fixed polynomial function for 20 trials, plotting the average performance and corresponding 95% confidence intervals in Figure 5. Figures 5a and 5b illustrate the performances of LCC and `LeTCC` for a low-degree polynomial and a small number of data points ($\deg(f) = 3$ and $K = 5$). In contrast, Figures 5c and 5d show performance for a higher-degree polynomial and a larger dataset ($\deg(f) = 15$ and $K = 10$). As shown in Figures 5a and 5b, LCC achieves exact results for $S \leq 7$. However, at larger values of $S$, as well as larger polynomial degree (as in Figures 5c and 5d), the proposed approach, without any parameter tuning, outperforms LCC in terms of both computational stability (lower variance) and recovery accuracy.

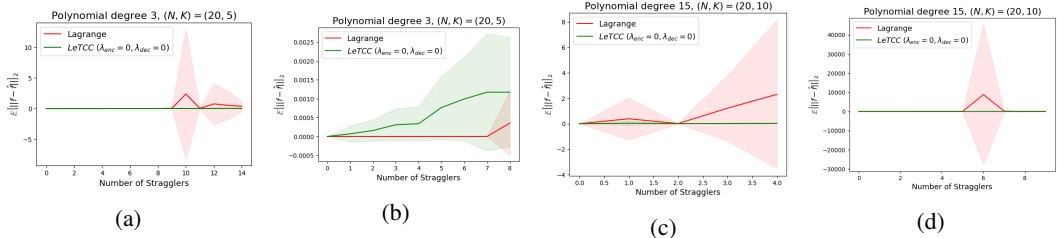

Figure 5: Average performance of `LeTCC` and Lagrange Coded Computing, with a 95% confidence interval. Plots (a) and (d) show the overall performance, while the zoomed-in subplots (b) and (c) highlight the performance for smaller range of stragglers.

## D.2 Computational complexity

Encoding and decoding complexities in `LCC` are $\mathcal{O}(N \cdot \log^2(K) \cdot \log\log(K) \cdot d)$ and $\mathcal{O}((N - S) \cdot \log^2((N - S)) \cdot \log\log((N - S)) \cdot m)$, respectively, where $d$ and $m$ are input and output dimensions of the computing function $f(\cdot)$, respectively [3]. In contrast, as mentioned before, for smoothing splines, the encoding and decoding process, which involves evaluation on new points and calculating the fitted coefficients, have the computational complexity of $\mathcal{O}(K.d)$ and $\mathcal{O}((N-s).m)$. Consequently, the computational complexity of the proposed scheme is less than `LCC`.

## E  Sensitivity Analysis

### E.1  Sensitivity to number of stragglers

The smoothing parameters for each model show low sensitivity to the number of stragglers (or worker nodes). To find the optimal smoothing parameter, we use cross-validation across different $\frac{S}{N}$ values. The following table presents the optimal smoothing parameters for selected numbers of stragglers for LeNet5 with $(N, K) = (100, 60)$ and RepVGG with $(N, K) = (60, 20)$, respectively. As shown in Table 3, the optimal values of $\lambda_e$ and $\lambda_d$ exhibit low sensitivity to the number of stragglers.

Table 3: Optimal smoothing parameters for different number of stragglers for LeNet and RepVGG architectures.

| (N, K) | LENET5 (100, 60) | | REPVGG (60, 20) | |
|---|---|---|---|---|
| S | $\lambda_e^*$ | $\lambda_d^*$ | $\lambda_e^*$ | $\lambda_d^*$ |
| 0 | $10^{-13}$ | $10^{-6}$ | $10^{-6}$ | $10^{-4}$ |
| 5 | $10^{-13}$ | $10^{-6}$ | $10^{-6}$ | $10^{-4}$ |
| 10 | $10^{-13}$ | $10^{-6}$ | $10^{-5}$ | $10^{-4}$ |
| 15 | $10^{-13}$ | $10^{-6}$ | $10^{-5}$ | $10^{-4}$ |
| 20 | $10^{-13}$ | $10^{-6}$ | $10^{-5}$ | $10^{-4}$ |
| 25 | $10^{-8}$ | $10^{-5}$ | $10^{-5}$ | $10^{-4}$ |
| 30 | $10^{-8}$ | $10^{-4}$ | $10^{-5}$ | $10^{-3}$ |
| 35 | $10^{-8}$ | $10^{-4}$ | $10^{-5}$ | $10^{-3}$ |

### E.2  Sensitivity to smoothing parameters

To assess the performance of the proposed scheme with respect to the smoothing parameters, we vary each parameter individually around its optimal point while holding the other parameter fixed at their optimal value. We then record the average percentage increase in RMSE relative to the RMSE at the

optimal point. Figure 6 presents these results for LeNet with $(N, K, S) = (100, 60, 20)$ (Figures 6a and 6b) and for RepVGG with $(N, K, S) = (60, 20, 35)$ (Figures 6c and 6d).

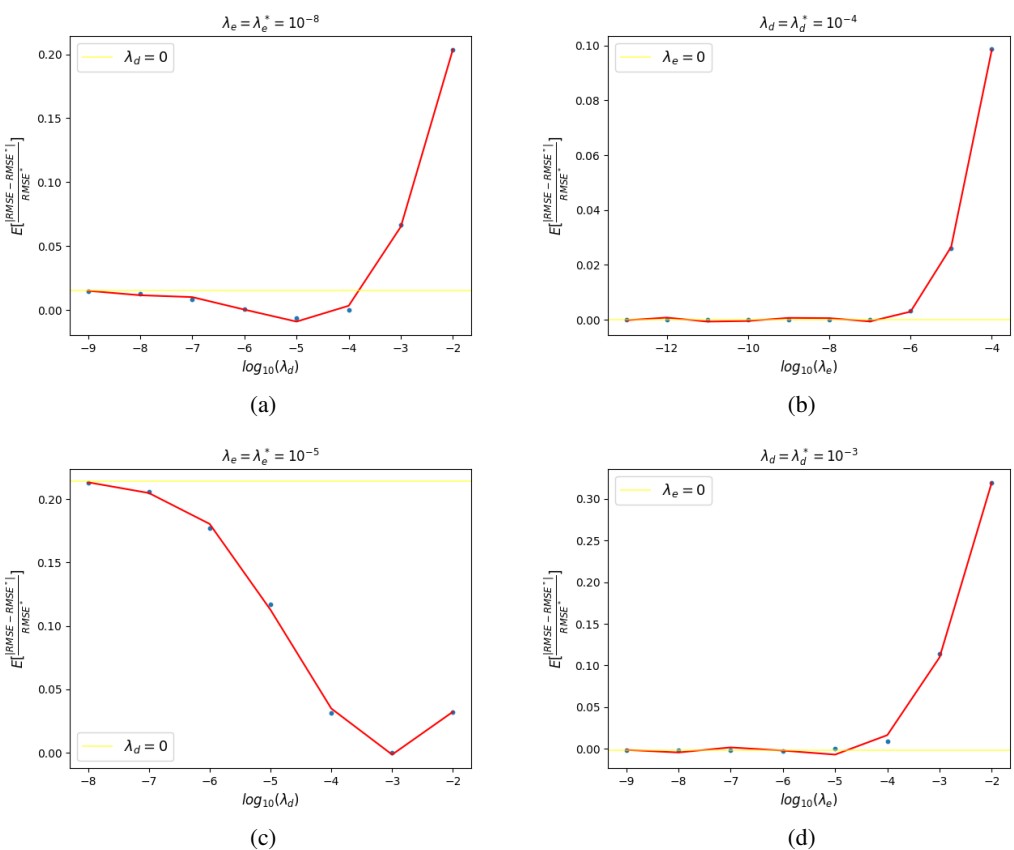

(a)  (b)

(c)  (d)

Figure 6: Sensitivity of LeTCC performance with respect to $\log_{10}(\lambda_d)$ and $\log_{10}(\lambda_e)$. The yellow line represents the performance when the variable smoothing parameter is set to zero.

As shown in Figure 6, the presence of more stragglers increases the sensitivity of LeTCC with respect to its smoothing parameter. However, even in a high-straggler regime, the RMSE increases by only around $3\%$ when the smoothing parameter deviates from its optimal value by a scale of 10.

## F  High-dimensional computing function

Let us consider more general cases where $f = [f_1, \ldots, f_m]$ is a vector-valued function, where each component function $f_j : \mathbb{R} \to \mathbb{R}$ is $q_j$-Lipschitz continuous. Based on (2), we have:

$$
\mathcal{R}(\hat{\mathbf{f}}) \leqslant \mathop{\mathbb{E}}_{\boldsymbol{\epsilon}, \mathcal{F} \sim F_{S,N}} \left[ \frac{2}{K} \sum_{k=1}^{K} \|\mathbf{u}_{\text{dec}}(\alpha_k) - \mathbf{f}(u_{\text{enc}}(\alpha_k))\|_2^2 \right] + \frac{2}{K} \sum_{k=1}^{K} \|\mathbf{f}(u_{\text{enc}}(\alpha_k)) - \mathbf{f}(x_k)\|_2^2.
$$

$$
\leqslant \mathop{\mathbb{E}}_{\boldsymbol{\epsilon}, \mathcal{F} \sim F_{S,N}} \left[ \frac{2}{K} \sum_{k=1}^{K} \sum_{j=1}^{m} \left( u_{\text{dec}_j}(\alpha_k) - f_j(u_{\text{enc}}(\alpha_k)) \right)_2^2 \right] + \frac{2 \sum_{j=1}^{m} q_j^2}{K} \sum_{k=1}^{K} \|u_{\text{enc}}(\alpha_k) - x_k\|_2^2
$$

$$
= \sum_{j=1}^{m} \mathop{\mathbb{E}}_{\boldsymbol{\epsilon}, \mathcal{F} \sim F_{S,N}} \left[ \frac{2}{K} \sum_{k=1}^{K} \left( u_{\text{dec}_j}(\alpha_k) - f_j(u_{\text{enc}}(\alpha_k)) \right)_2^2 \right] + \frac{2 \sum_{j=1}^{m} q_j^2}{K} \sum_{k=1}^{K} \|u_{\text{enc}}(\alpha_k) - x_k\|_2^2
$$

Let us define the following objective for the decoder function:

$$
\mathbf{u}_{\text{dec}}^{\star} = \operatorname*{argmin}_{\mathbf{u} \in \mathcal{H}^2(\Omega; \mathbb{R}^M)} \frac{1}{|\mathcal{F}|} \sum_{v \in \mathcal{F}} \|\mathbf{u}(\beta_v) - \mathbf{f}(u_{\text{enc}}(\beta_v))\|_2^2 + \sum_{j=1}^{m} \lambda_d \int_{\Omega} \left( u_j''(t) \right)^2 dt. \tag{131}
$$

The solution to (131), denoted as $\mathbf{u}_{\text{dec}}^{\star}$, is a vector-valued function, where each component $u_{\text{dec}_j}(\cdot)$ is a smoothing spline function fitted to the data points $\{(\beta_v, f_j(u_{\text{enc}}(\beta_v)))\}_{v \in \mathcal{F}}$. As a result, By defining $q = \sqrt{\sum_{j=1}^{m} q_j^2}$ and scaling up all upper bounds for $\mathcal{L}_{\text{dec}}$ by a factor of $m$, all previous results and theorems seamlessly extend to high-dimensional computing functions.

## G    Coded data points

Figures 7b and 7c display coded samples generated by BACC and LeTCC, respectively, derived from the same initial data points depicted in Figure 7a. These samples are presented for the MNIST dataset with parameters $(N, K) = (70, 30)$. From the figures, it is apparent (Specifically in paired ones that are shown with the same color) that while both schemes' coded samples are a weighted combination of multiple initial samples, BACC's coded samples exhibit high-frequency noise. This observation suggests that LeTCC regression functions produce more refined coded samples without any disruptive noise.

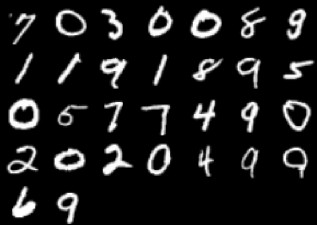

(a) Initial inputs

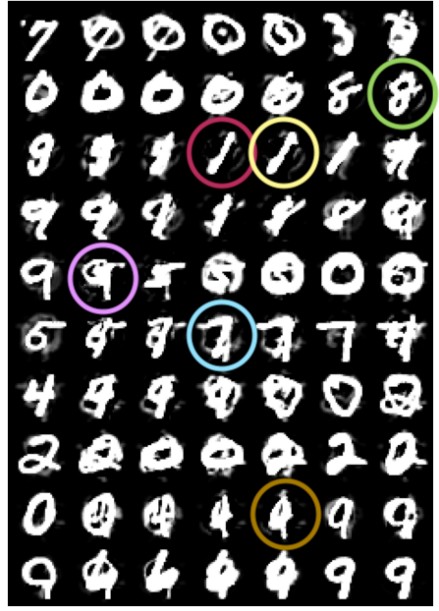

(b) BACC coded samples

(c) LeTCC coded samples

Figure 7: Comparison of coded samples between BACC and LeTCC frameworks. Figure 7a represents the initial data points $\{\mathbf{x}_k\}_{k=1}^{K}$ for $K = 30$. Figures 7b and 7c display $N = 70$ coded samples $\{\tilde{\mathbf{x}}_n\}_{n=1}^{N}$ from BACC and LeTCC, respectively. Samples with clear differences are highlighted with the same color.

