# OpenReview forum: "Coded Computing for Resilient Distributed Computing: A Learning-Theoretic Framework"
_NeurIPS.cc/2024/Conference — NeurIPS 2024 poster_

### Official Review · Reviewer_a2Af · 2024-07-09

**Soundness:** 3
**Presentation:** 3
**Contribution:** 2
**Rating:** 4
**Confidence:** 3

**Summary:**

This paper focuses on coded computing for machine learning and derives loss-minimizing encoding and decoding functions.

**Strengths:**

Please see the “Questions” section.

**Weaknesses:**

Please see the “Questions” section.

**Questions:**

My review is as follows:

Major:

The biggest thing that confused me when reading the paper was the use case for this kind of method. As far as I understand, this method is specifically designed for inference. Inference of computer vision models such as the ones used in the experiments of this paper (VGG, VIT) is actually very fast in practice. Even mobile phones can run inference on these models within a few tens or hundreds of milliseconds. What application would require to run many inferences for these models in parallel in a timely manner?

I understand that the results are interesting from a theoretical point of view. It’s nice that encoding and decoding functions can be derived theoretically. But from a practical perspective, I cannot think of a scenario where one would need to make their inference straggler-resilient because it is typically already very low latency. (Furthermore, if an even lower latency is needed, then quantization is usually the direction to explore.) Please let me know if I’m missing something here. Perhaps, straggler resilient computing makes more sense for large-scale computing (e.g. large-model training) where you have a large-scale computation that is distributed onto a compute cluster.

Minor:

Some of the related work on coded computing has the property that if a given number of nodes return their solution, the exact output can be recovered and if not, an approximate solution can be generated. As far as I understand, the proposed method cannot recover the exact result even if all nodes return their result successfully. Please correct me if I’m wrong about this point. I wonder if this method could be modified to have the exact recovery property.

**Limitations:**

Yes.

---

> ### Author Rebuttal · Authors · 2024-08-07
>
> We express our gratitude to the reviewer for their valuable comments and feedback. In the above, we have offered a general response to the reviewers, addressing some of their common concerns. Here, we provide additional responses to the remaining questions raised.
>
> > Major:
> >The biggest thing that confused me when reading the paper was the use case for this kind of method. As far as I understand, this method is specifically designed for inference. Inference of computer vision models such as the ones used in the experiments of this paper (VGG, VIT) is actually very fast in practice. Even mobile phones can run inference on these models within a few tens or hundreds of milliseconds. What application would require to run many inferences for these models in parallel in a timely manner?
> > I understand that the results are interesting from a theoretical point of view. It’s nice that encoding and decoding functions can be derived theoretically. But from a practical perspective, I cannot think of a scenario where one would need to make their inference straggler-resilient because it is typically already very low latency. (Furthermore, if an even lower latency is needed, then quantization is usually the direction to explore.) Please let me know if I’m missing something here. Perhaps, straggler resilient computing makes more sense for large-scale computing (e.g. large-model training) where you have a large-scale computation that is distributed onto a compute cluster.
>
> **Answer:** We appreciate the reviewer’s valuable feedback. We would like to answer this as follows:
>
> 1. Machine Learning as a Service (MLaS) is a new paradigm where resource-constrained clients outsource their computationally expensive tasks to powerful clouds such as Amazon, Microsoft, and Google [1]. Consequently, prediction serving systems in these powerful clouds host complex machine learning models and respond to a vast number of inference queries worldwide with low latency [1]. However, the limited number of worker nodes, the large volume of inference queries, and the ever-increasing complexity of models make this task even more challenging. In this context, addressing the straggler issue and making the entire system straggler-resistant is crucial. As shown in [2], stragglers are inevitable even in simple tasks like matrix multiplication. Therefore, straggler resiliency is a vital feature of any prediction serving system.
>
> 2. Inference is just one example of the proposed framework's applications. As mentioned in the paper, the framework is designed for general computing functions. Notably, inference is a crucial task, as it has been used as a benchmark in other works, such as [1].
> By utilizing the gradient of a model's loss with respect to its parameters as the computing function, $f(x) =\nabla L(x; \theta)$, where $L$ is the loss of a large pre-trained neural network like large language models, the proposed framework can be applied to fine-tuning large neural networks as well. We consider this case as a potential direction for our future work.
>
> [1] Soleymani, M., et al. "ApproxIFER: A model-agnostic approach to resilient and robust prediction serving systems." AAAI 2022.
>
> [2] Gupta, V., et al. "Oversketch: Approximate matrix multiplication for the cloud." Big Data 2018.
>
> > Minor:
> Some of the related work on coded computing has the property that if a given number of nodes return their solution, the exact output can be recovered and if not, an approximate solution can be generated. As far as I understand, the proposed method cannot recover the exact result even if all nodes return their result successfully. Please correct me if I’m wrong about this point. I wonder if this method could be modified to have the exact recovery property.
>
> **Answer:**  We appreciate the reviewer’s valuable feedback. To the best of our knowledge, for **general computation**, there is no solution that has the property of exact recovery for a small number of stragglers and approximate results for a larger straggler. On the other hand, for specific structured computations such as polynomial computation, Lagrange coded computing [1] can give us exact output if $(K-1)\cdot \operatorname{deg}(f) + S < N$. However, the decoding algorithm of Lagrange coded computing does not work for the cases where the above condition does not hold. If we use a heuristic solution in which we fit a lower degree polynomial to the existing solution, the approximation is not acceptable (see Figure 3 in the attachment).
>
> Additionally, if the function that we want to compute belongs to the space of functions generated through smoothing spline basis (refer to Equation (5) in the paper), then exact recovery is also feasible in our proposed scheme with appropriate hyper-parameters and appropriate condition on the number of stragglers.
>
> [1] Yu, Q., et al. "Lagrange coded computing: Optimal design for resiliency, security, and privacy." PMLR 2019.

---

> ### Author Response · Authors · 2024-08-12
> **Any feedback on the rebuttal?**
>
> Dear Reviewer a2Af,
>
> As we approach the conclusion of the author-reviewer discussion phase, we wish to gently remind you that we remain available to address any additional questions or concerns you may have before finalizing your score.
>
> Best regards,
> The Authors

---

> > ### Comment · Reviewer_a2Af · 2024-08-12
> >
> > Thank you for the rebuttal. I still think that if the stragglers were a big issue in inference serving, then the proposed method would be an interesting solution. However, I don't really see the straggling nodes to be a real issue in practice for inference serving.
> >
> > I agree with the authors' point that the model complexity keeps growing. We should keep in mind that the largest models are typically genAI models such as LLMs or image generation models (e.g. stable diffusion). LLMs generate tokens autoregressively and stable diffusion has an iterative denoising component.

---

> ### Author Response · Authors · 2024-08-14
>
> Thank you for your valuable feedback.
>
> The importance of stragglers in distributed computing was raised in a seminal paper by Google [1]. The proposed framework introduces a straggler-resistant scheme for general computing, which is not limited to inference. For numerical evaluation, we chose machine learning inference as a benchmark, as it has been used in related papers such as [2].
>
> [1] Dean, J., et. al, "The tail at scale." Communications of the ACM 2013
>
> [2] Soleymani, M., et al, "ApproxIFER: A model-agnostic approach to resilient and robust prediction serving systems." AAAI 2022.

---

### Official Review · Reviewer_SdcZ · 2024-07-11

**Soundness:** 3
**Presentation:** 4
**Contribution:** 2
**Rating:** 7
**Confidence:** 2

**Summary:**

The authors consider the problem of improving the reliability of distributed computing architectures by encoding the input data before it is processed by worker machines such that a good approximation of the desired output can be reconstructed using only a subset of the workers’ outputs. They utilize learning theory to determine a regularized objective function for the decoder as well as a loss function for the overall system, and provide solutions for the encoder and decoder which optimize their derived upper bounds on the loss function. The objective function is based on kernel ridge regression, which leads to a second-order smoothing spline solution for the decoder. Both the noiseless and the noisy cases are considered, and the proposed method is compared with the existing approach of Berrut coded computing in terms of its convergence rate in the noiseless setting and its empirical performance. To evaluate the empirical performance, distributed inference is performed for deep learning models of varying sizes and tasks with varying sizes for the output vector. The experimental results show that LETCC consistently achieves a lower estimation error on the model outputs.

**Strengths:**

# Originality
The main contribution which differentiates the LeTCC method from prior works like BACC appears to be the introduction of regularization which causes the optimal solution to be based on smoothing splines rather than interpolation. Existing papers which have considered optimizing cost functions only focus on learning how to generate parity data points. Overall, this paper brings a novel method to the table and adequately cites related works.
# Quality
Although I lack experience with some of the mathematical tools used, to my knowledge there is nothing incorrect about the technical results. The experimental methodology seems to allow for a fair comparison between LeTCC and BACC for the use case of image classification with a deep learning model. In terms of reconstruction accuracy, the experimental results validate that LeTCC is superior to BACC, in some cases by a significant margin.
# Clarity
The main content of the paper is laid out clearly with some small typos but no major issues.
# Significance
The method for coded computing introduced in this paper represents an advancement in the level of accuracy that can be achieved when the function applied by the worker nodes is a complex deep learning model. The authors show that this advancement can be achieved by adapting the encoder and decoder to the function through the tuning of the regularization weight, which is an interesting idea.

**Weaknesses:**

# Originality
This paper distinguishes itself sufficiently from existing works, so I see no issues in terms of originality.
# Quality
LeTCC is proven to have a reconstruction error that scales better than BACC in terms of the number of worker nodes, but its scaling in terms of the number of stragglers is not directly compared. Based on Theorem 9 in the BACC paper, it appears that the max error along a single dimensions scales as $S^2$. The mean-squared error (MSE) for LeTCC scales as $S^4$, but since the max norm is upper bounded by the Euclidean norm it is unclear how to compare these results. It would be nice to compare the theoretical dependence on the number of stragglers, especially since the experimental results are mixed (in two cases the difference in MSE shrinks as $S$ increases, and in the other case the MSE grows as $S$ increases).
# Clarity
The authors should explicitly state how many trials were averaged over to produce the plots in figure 3 to help readers interpret and/or reproduce the results.

I identified the following typos in the paper:
- Line 502: There is a “loc” subscript missing
- Line 49: The sentence starting on this line is grammatically incorrect, I presume that the second comma is supposed to be replaced with the word “is”
- Figure 3: The title of the top-left plot seems to have the values of N and K reversed

# Significance
Without more details on the hyperparameter optimization process and the relative computational complexity of fitting the encoder and decoder functions as compared to those of BACC, it is hard to say whether LeTCC is more practically advantageous to use than BACC. It could be that the overhead introduced by hyperparameter tuning outweighs the benefit from reduced reconstruction error in some cases. It would have been helpful to see how tolerant LeTCC is to changes in the smoothing parameters, i.e. how much variation can be allowed while still outperforming BACC. One can also see from Figure 3 that a large difference in the mean-squared error does not consistently translate to a large difference in relative accuracy (at least when deep models are used), which makes it less likely that LeTCC’s performance benefits would be worth the additional overhead. As it currently stands, these issues reduce the practical value of the contribution made by this paper.

**Questions:**

According to Theorem 4, the mean-square error of LeTCC has a much worse scaling in terms of the number of stragglers in the noiseless setting than in the noisy setting. Does this originate from a sacrifice made to improve the scaling with respect to the number of worker nodes, or is there some other cause?

**Limitations:**

The authors are upfront about the limited scope of their work. However, as mentioned in the weaknesses section, the extra overhead introduced by the need for hyperparameter tuning is something that I would have liked to have seen addressed in the paper. As mentioned in the weaknesses section, the comparison of error convergence rates with BACC is also limited.

---

> ### Author Rebuttal · Authors · 2024-08-07
>
> We express our gratitude to the reviewer for their valuable comments and feedback. In the above, we have offered a general response to the reviewers, addressing some of their common concerns. Here, we provide additional responses to the remaining questions raised.
>
> > LeTCC is proven to have a reconstruction error that scales better than BACC in terms of the number of worker nodes, but its scaling in terms of the number of stragglers is not directly compared. Based on Theorem 9 in the BACC paper, it appears that the max error along a single dimensions scales as S^2 The mean-squared error (MSE) for LeTCC scales as S^4 but since the max norm is upper bounded by the Euclidean norm it is unclear how to compare these results. It would be nice to compare the theoretical dependence on the number of stragglers, especially since the experimental results are mixed (in two cases the difference in MSE shrinks as S increases, and in the other case the MSE grows as  S increases).
>
>
> **Answer:**  We thank the reviewer for the detailed question. Note that our proposed solution achieves a factor of $S^4$ for the **squared** $\ell_2$ norm of the error, compared to the factor of $S^2$  for the $\ell_\infty$ norm of error (equivalently $S^4$ factor for  **squared** $\ell_\infty$ norm) in the BACC paper. Thus, if we compare these two schemes since $\ell_2$ norm is upper bounded by $\ell_{\infty}$ norm, both of them have the factor of $S^4$ for the upper bound of the $\ell_2$ norm of the error. Therefore, we cannot conclude that BACC performs better than LeTCC in terms of the scale of the error as a function of $S$.
>
> > The authors should explicitly state how many trials were averaged over to produce the plots in figure 3 to help readers interpret and/or reproduce the results.
>
> **Answer:** We have provided information on the statistical properties of our experiments, including confidence intervals, in the general rebuttal and Figures 1 and 2 of the attached PDF. For a comprehensive overview of our experimental results and statistics, please see the general rebuttal and the attached file.
>
>
> >I identified the following typos in the paper:
>
> >> - Line 502: There is a “loc” subscript missing
>
> >>- Line 49: The sentence starting on this line is grammatically incorrect, I presume that the second comma is supposed to be replaced with the word “is”
>
> >> - Figure 3: The title of the top-left plot seems to have the values of N and K reversed
>
>
>
> **Answer:** Thank you for pointing out the typos in our paper. We appreciate your attention to detail and have corrected them.
>
> > Without more details on the hyperparameter optimization process and the relative computational complexity of fitting the encoder and decoder functions as compared to those of BACC, it is hard to say whether LeTCC is more practically advantageous to use than BACC. It could be that the overhead introduced by hyperparameter tuning outweighs the benefit from reduced reconstruction error in some cases. It would have been helpful to see how tolerant LeTCC is to changes in the smoothing parameters, i.e. how much variation can be allowed while still outperforming BACC. One can also see from Figure 3 that a large difference in the mean-squared error does not consistently translate to a large difference in relative accuracy (at least when deep models are used), which makes it less likely that LeTCC’s performance benefits would be worth the additional overhead. As it currently stands, these issues reduce the practical value of the contribution made by this paper.
>
> **Answer:** Please refer to the general rebuttal and attached PDF for comprehensive information on the experiments, including smoothing parameter sensitivity, experimental statistics, and a comparison of the computational complexity of the proposed model.
>
> > According to Theorem 4, the mean-square error of LeTCC has a much worse scaling in terms of the number of stragglers in the noiseless setting than in the noisy setting. Does this originate from a sacrifice made to improve the scaling with respect to the number of worker nodes, or is there some other cause?
>
> **Answer:** Based on Theorem 4, the error convergence rates of noiseless and noisy settings are $\mathcal{O}(S\cdot(\frac{S}{N})^3)$ and $\mathcal{O}(S\cdot(\frac{S}{N})^\frac{3}{5})$, respectively. We note that, for $S<N$, $S\cdot(\frac{S}{N})^3 < S\cdot(\frac{S}{N})^\frac{3}{5}$ and thus, as we expect, the upper bound on the error of the noiseless cases scales better compare to that of noisy ones. It is important to note that considering the variation of these bounds as a function of $S$ without considering the effect of $N$ may lead to a misleading conclusion
>
> >Limitations: The authors are upfront about the limited scope of their work. However, as mentioned in the weaknesses section, the extra overhead introduced by the need for hyperparameter tuning is something that I would have liked to have seen addressed in the paper. As mentioned in the weaknesses section, the comparison of error convergence rates with BACC is also limited.
>
> **Answer:** We thank the reviewer for pointing out the limitations of out work. We have already addressed some of them in the general rebuttal and we will include them as a separate section in the revised paper. Also we decided to add a full section to the revised version of the paper to discuss the limitations in a more structured way in the paper.

---

> > ### Comment · Reviewer_SdcZ · 2024-08-12
> >
> > Thank you for your response. Most of my concerns have been addressed, although the nature of the sensitivity experiment which the authors conducted does not exactly explain how bad the performance drop will be if the value of the smoothing parameters is different from the optimal values by a given amount. I raise my score to 7.

---

> > > ### Author Response · Authors · 2024-08-14
> > >
> > > Thank you for you insightful comments and feedback. We will address the complete sensitivity analysis in the revised version.

---

### Official Review · Reviewer_JpiK · 2024-07-12

**Soundness:** 3
**Presentation:** 2
**Contribution:** 2
**Rating:** 6
**Confidence:** 3

**Summary:**

This work proposes a learning theory-based novel framework for coded computing with a focus on distributed machine learning applications. The proposed method sends mixtures of input samples to the worker nodes that compute the desired results on the mixtures. An encoder and a decoder functions are fitted at the master node using input samples and the result from workers, respectively. Finally, the decoder function can be used to estimate the output of computing function for the input samples. It is shown that the loss function (divergence of the estimated output with the true output of computing functions) is upper bounded by the generalization error of the decoder regression and training error of the encoder regression under some conditions. Experimental evaluations are provided to show the efficacy of the proposed method over the state-of-the-art baseline.

**Strengths:**

1. The proposed method leverages learning theory for coded computing which seems novel and interesting.
2. Rigorous theoretical analysis is provided including convergence rate derivation and recoverability analysis.

**Weaknesses:**

1. The experimental section does not look comprehensive enough for the following reasons:
   (a) Only one baseline is considered.
   (b) Authors acknowledge in Section 3 that the computational efficiency of encoder and decoders is a crucial factor. Yet, this was not reported in the numerical section.

2. The computing functions (which are generally pre-trained neural networks) are evaluated on (probably) mixtures of input samples at the worker nodes. Those neural networks were trained on the data distribution. However, the mixture of input samples might not fall on the support of data distribution. Although for images, this does not seem to be an issue, there could be unexpected behavior in general.

**Questions:**

1. Could the authors provide comparision with additional baselines?
2. Could the authors report the computational resources needed for fitting and evaluating the encoder and decoder?
3. Could the authors provide a discussion for the weaknesses number 2 explained above? Also, does this framework work (or can it be modified to work) for computing functions that take discrete inputs?

**Limitations:**

Yes, but it is spread out throughout the manuscript. A separate limitation section/paragraph would be helpful for the readers.

---

> ### Author Rebuttal · Authors · 2024-08-07
>
> We express our gratitude to the reviewer for their valuable comments and feedback. In the above, we have offered a general response to the reviewers, addressing some of their common concerns. Here, we provide additional responses to the remaining questions raised.
>
> > The experimental section does not look comprehensive enough for the following reasons: (a) Only one baseline is considered.
> (b) Authors acknowledge in Section 3 that the computational efficiency of encoder and decoders is a crucial factor. Yet, this was not reported in the numerical section.
>
> >>Could the authors provide comparison with additional baselines?
>
> >>Could the authors report the computational resources needed for fitting and evaluating the encoder and decoder?
>
>
> **Answer:** We thank the reviewer for their constructive comment. Although there is only one baseline for general computing approximation, we compared our proposed scheme with Lagrange Coded Computing for polynomial computation. Additionally, we have analyzed the computational complexity of our scheme and compared it with BACC and Lagrange. Please refer to our general response for a detailed discussion.
>
> > The computing functions (which are generally pre-trained neural networks) are evaluated on (probably) mixtures of input samples at the worker nodes. Those neural networks were trained on the data distribution. However, the mixture of input samples might not fall on the support of data distribution. Although for images, this does not seem to be an issue, there could be unexpected behavior in general.
>
> >>Also, does this framework work (or can it be modified to work) for computing functions that take discrete inputs?
>
>
> **Answer:** We appreciate the reviewer's thoughtful and detailed comment. Firstly, as established in Theorems 1-3, our results are predicated on the assumption of computing function smoothness, which is guaranteed by bounding the maximum norm of its first and second derivatives. Consequently, our framework ensures high recovery accuracy only when the computing function exhibits smoothness. If this smoothness assumption is not met, our framework cannot guarantee accurate recovery.
>
> Secondly, based on the results of Theorems 1-3 and Corollary 2, we have shown that the optimal encoder is the smoothing spline. It is shown that the smoothing spline operator is asymptotically equivalent to a kernel regression estimator with a Silverman kernel, whose local bandwidth is $\lambda^{\frac{1}{4}}q(t)^{\frac{-1}{4}}$, where $\lambda$ is the smoothing parameter and $q(t)$ is the probability density function of the input data points [1]. This leads to the key observation:
>
>
> * In the smoothing spline, the bandwidth (i.e., the number of data points effectively contributing to the input data of a worker node) and their corresponding weights depend on the input data distribution as well as the smoothing parameter. This property makes our approach more generalizable to various data distributions. If the data distribution is such that a linear combination of input data points may be inappropriate, we can control the number of points involved in the linear combination and their weights by choosing the right smoothing parameter. This ensures that the output of the linear combination remains reasonably close to the data distribution on which the model was trained.
>
> In contrast, the Berrut approach has a fixed, bounded bandwidth due to the $\frac{1}{z-\alpha}$ factor in the numerator of the data points coefficient
> $$u_{enc}(z)=\sum_{i=0}^{K-1} \frac{\frac{(-1)^i}{\left(z-\alpha_i\right)}}{\sum_{j=0}^{K-1} \frac{(-1)^j}{\left(z-\alpha_j\right)}} \mathbf{x}_i$$
> Therefore, the problem that the reviewer mentioned will affect BACC more. This observation suggests another perspective on why the proposed solution outperforms BACC.
>
>
> Finally, in the case of discreet input, if the model exhibits "smoothness" (i.e., it accepts continuous input and has bounded first and second derivatives, ensuring that small changes in the input result in minimal output changes), our proposed scheme will be effective.
>
> [1] Silverman, B.W. "Spline smoothing: the equivalent variable kernel method." The Annals of Statistics, 1984
>
> > Limitations: Yes, but it is spread out throughout the manuscript. A separate limitation section/paragraph would be helpful for the readers.
>
> **Answer:** We thank reviewer for the helpful comment. We will add a dedicated section to the  limitations of our work in next version.

---

> > ### Comment · Reviewer_JpiK · 2024-08-09
> >
> > Thank you for the additional experiments, discussion about the baselines and question 3. The authors have successfully addressed my comments.
> > Therefore, I raise my score to 6.

---

> > > ### Author Response · Authors · 2024-08-14
> > >
> > > Thank you for your insightful comments.

---

### Official Review · Reviewer_Uhit · 2024-07-12

**Soundness:** 3
**Presentation:** 3
**Contribution:** 3
**Rating:** 6
**Confidence:** 3

**Summary:**

The paper deals with coded distributed computing. I need to note that it is very popular research area now with a vast number of papers. But the authors are right when mention that the majority of papers utilizes standard algebraic codes (such as Reed-Solomon codes). The main problem of such approach is that these codes are designed over finite fields, which is not the case for machine learning tasks. The improvements utilize real or complex valued codes (e.g. Reed-Solomon codes over real or complex fields) but they face with the problems of accuracy (Vandermonde matrices have bad condition numbers). Moreover such approaches work only for some particular functions, e.g. polynomial ones. In this paper the authors propose a framework, which outperforms the Berrut approximate coded computing (BACC), which is the state-of-the-art coded computing scheme for general computing.

**Strengths:**

The main strong point are as follows:
- new framework foundation for coded computing, new loss function and its upper bound by decomposition
- theoretical analysis and guarantees. Under some assumptions the authors find the optimum encoding and decoding functions and characterize the convergence rate for the expected loss in both noise-free and noisy computations.
- Numerical analysis, that shows the new framework to outperforms the Berrut approximate coded computing (BACC), which is the state-of-the-art coded computing scheme for general computing.

**Weaknesses:**

I list the main weaknesses below:
- «We develop a new foundation for coded computing, based on the theory of learning, rather than the theory of coding». I would not be so categorical. Coding is a method when you add and the utilize the redundancy to deal with errors and erasures. The main advantage of your approach is that you are not using standard finite field codes.
- How the optimal (under some assumptions) encoder and decoder functions in section 4 are related to the functions utilized for numerical experiments (DNNs)? Is it possible to analyse the performance under optimal encoder and decoder and compare it to the results from Section 5?
- You made the comparison to BACC, which is a framework for general computing. Could you please make a comparison for some particular (e.g. polynomial) problems? I just wonder if proposed general approach is competitive with e.g. Lagrange computing and what you should pay for universality. So my claim is that more scenarios and computing problems should be checked to understand the limitations and applicability of your method.
- How flexible is your approach if the system parameters change (the number of worker nodes or the number of stragglers).
- Figure 3 seems to suffer from the lack of statistics (you need more experiments).
- You mentioned Byzantine case. It is known to be a problem for the codes over the real field (in case of finite field the approaches are well-developed). It would be beneficial to briefly explain how you plan to deal with this problem.

**Questions:**

See weaknesses

**Limitations:**

Limitations are well described.

---

> ### Author Rebuttal · Authors · 2024-08-07
>
> We express our gratitude to the reviewer for their valuable comments and feedback. In the above, we have offered a general response to the reviewers, addressing some of their common concerns. Here, we provide additional responses to the remaining questions raised.
>
> > «We develop a new foundation for coded computing, based on the theory of learning
> rather than the theory of coding». I would not be so categorical. Coding is a method when you add and the utilize the redundancy to deal with errors and erasures. The main advantage of your approach is that you are not using standard finite field codes.
>
> **Answer:** We appreciate the reviewer’s valuable feedback. We would like to clarify our first contribution. As we mentioned in the objective box (line 70):
> > The main objective of this paper is to develop a new foundation for coded
> computing, **not solely based on coding theory, but also grounded in learning theory**
>
> as well as abstract (lines 8 and 9):
> > we propose a novel foundation for coded computing, **integrating the principles of learning theory**, and developing a new framework that seamlessly adapts with machine learning applications.
>
> Therefore, what we mean by "based on the learning theory" is basically integrating a learning theoretic mindset, on top of coding theory, into the whole framework, including (i) considering the whole system as an end-to-end system, (ii) defining its corresponding loss function and (iii) deriving optimum encoder and decoder function using theories from kernel regression. It does not mean that we are not using coding and learning theoretical approaches instead. Conventional coded computing did not consider the whole system view and was solely based on algebraic coding theory, originally developed for communication theory, which makes them not well generalizable to machine learning applications.
>
> We agree that this sentence in Line 86 could cause misunderstanding, and we will change it accordingly.
>
> > How the optimal (under some assumptions) encoder and decoder functions in section 4 are related to the functions utilized for numerical experiments (DNNs)? Is it possible to analyze the performance under optimal encoder and decoder and compare it to the results from Section 5?
>
> **Answer:** In fact, we do use the optimal encoder and decoder functions in the numerical experiments. We proved in Corollary 4 (lines 236-240) that the optimal encoder function is the smoothing spline function, just like the decoder function. As a result, we use smoothing spline functions for our experiments in both encoder and decoder with smoothing parameters $\lambda_{enc}$ and $\lambda_{dec}$ respectively. For more clarification, we will mention this fact directly in our experimental setup section (Section 5).
>
> > You made the comparison to BACC, which is a framework for general computing. Could you please make a comparison for some particular (e.g. polynomial) problems? I just wonder if proposed general approach is competitive with e.g. Lagrange computing and what you should pay for universality. So my claim is that more scenarios and computing problems should be checked to understand the limitations and applicability of your method.
>
> **Answer:** We have provided further comparisons with Lagrange Coded Computing in the general rebuttal, as well as in Figure 3 of the attached PDF.
>
> > How flexible is your approach if the system parameters change (the number of worker nodes or the number of stragglers).
>
> **Answer:** We have included a discussion on the sensitivity of the framework's hyper-parameters ($\lambda_{enc}$ and $\lambda_{dec}$) to the number of stragglers in the general rebuttal.
>
> > Figure 3 seems to suffer from the lack of statistics (you need more experiments).
>
> **Answer:** We have included a detailed description of our experiments and added confidence intervals to the plots. Please refer to the general rebuttal and Figures 1 and 2 in the attached PDF for a comprehensive discussion.
>
> > You mentioned the Byzantine case. It is known to be a problem for the codes over the real field (in the case of the finite field, the approaches are well-developed). It would be beneficial to briefly explain how you plan to deal with this problem.
>
> **Answer:** We appreciate the reviewer's insightful comment. While analyzing the scheme in the presence of Byzantine faults is beyond the scope of this paper, which primarily focuses on straggler resiliency, we acknowledge the importance of this aspect. Similar to [1], which enhances Berrut coded computing with Byzantine robustness using the Berlekamp-Welch (BW) decoding algorithm for Reed-Solomon codes [2], we plan to explore the same approach to incorporate Byzantine robustness into our framework. Still, it needs significant work to design an effective error correction algorithm and develop a theoretical guarantee tailored to our framework.
>
>
> [1] Soleymani, M., et al. "ApproxIFER: A model-agnostic approach to resilient and robust prediction serving systems." AAAI 2022.
> [2] Blahut, R.E., "Algebraic codes on lines, planes, and curves: an engineering approach." Cambridge University Press 2008.

---

### Author Rebuttal · Authors · 2024-08-07

#  General Response to Reviewers

We appreciate the reviewers' constructive feedback. Here, we provide a general response to their common questions.

**Experiments.** In our revised version, we present a more comprehensive evaluation by incorporating the statistical properties of our experiments. Specifically, for each number of stragglers $S$, we evaluate LeTCC and BACC using the same input data points $\mathbf{x}_1, \dots, \mathbf{x}_k$ and repeat the experiment $20$ times with different sets of randomly chosen input data. We then plot the average result with a 95% confidence interval, providing a clearer picture of the performance and variability of each method. Figures 1 and 2 in the attached PDF display the average performance at a 95% confidence interval. Figure 1 shows a visible performance gain in both the RelAcc and MSE when $\frac{N}{K}$ is small (this is the practically important case, where the system is not over-designed with highly coded redundant computing). On the other hand, when the system is over-designed with a smaller $\frac{N}{K}$ ratio, then LeTCC shows a minor improvement compared to BACC, specifically in the relative accuracy.

We will include the new experiments (the ones depicted in Figure 1 of the attached PDF) in our next version.

**Sensitivity Analysis.** The smoothing parameters for each model exhibit low sensitivity to the number of stragglers (or worker nodes). To determine the optimal smoothing parameter, we employ cross-validation for different values $\frac{S}{N}$. The following tables display the optimal smoothing parameters for some numbers of stragglers for LeNet5 with $(N, K) = (100, 60)$ and RepVGG with $(N, K) = (60, 20)$, respectively. We will include this discussion in the revised version of the paper. As we can see, the optimum values of $\lambda^*_{enc}$ and $\lambda^*_{dec}$ are not sensitive to the number of stragglers.


|#Stragglers|$\lambda^*_{enc}$|$\lambda^*_{dec}$|
|-|-|-
|0|$10^{-13}$|$10^{-6}$
|5|$10^{-13}$|$10^{-6}$
|10|$10^{-13}$|$10^{-6}$
|15|$10^{-13}$|$10^{-6}$
|20|$10^{-13}$|$10^{-6}$
|25|$10^{-8}$|$10^{-5}$
|30|$10^{-8}$|$10^{-4}$
|35|$10^{-8}$|$10^{-4}$

|#Stragglers|$\lambda^*_{enc}$|$\lambda^*_{dec}$
|-|-|-
|0|$10^{-6}$|$10^{-4}$|
|5|$10^{-6}$|$10^{-4}$|
|10|$10^{-5}$|$10^{-4}$|
|15|$10^{-5}$|$10^{-4}$|
|20|$10^{-5}$|$10^{-4}$|
|25|$10^{-5}$|$10^{-4}$|
|30|$10^{-5}$|$10^{-3}$|
|35|$10^{-5}$|$10^{-3}$|

**Other Baselines.** Note that the **only** existing coded computing scheme for general functions is Berrut coded computing [1], which we have compared with our proposed scheme. Other schemes handle specific computations like polynomial functions [2] and matrix multiplication [3].

As suggested by the reviewers, we compare our proposed scheme with Lagrange coded computing [2], designed for polynomial computations:

* **Accuracy of function approximation**: Lagrange coded computing is only applicable to polynomial computing functions [2]. Also, the number of servers required to recover must be at least $(K-1)\times \text{deg}(f)+S+1$ worker nodes [1, 2]; otherwise, the master node cannot recover anything. Finally, the Lagrange coded computing is designed for computation over finite fields, and it faces serious instability problems in computation over real numbers when $(K-1)\times \text{deg}(f)$ is around $10$ or more [1, 4].
We compare the proposed framework (LeTCC), and the Lagrange coded computing in Figure 3 in the attached document. Recall that if $N < (K-1)\times \text{deg}(f)+S+1$, Lagrange coded computing does not work. Still, to push the application of Lagrange coded computing to those cases, we can fit a lower degree polynomial to the existing workers' results to get approximate results. We run LeTCC and Lagrange coded computing for the same set of input data and fixed polynomial function 20 times and plot the average performance with a 95% confidence interval in Figures 3a, 3b and 3c, 3d. Figures 3a and 3b show performances of Lagrange and LeTCC for the case where the degree of the polynomial and the number of data points are small ($\text{deg}(f)=3$ and $K=5$), while Figures 3c and 3d show the performance for larger polynomial degree and number of data points($deg(f)=15$ and $K=10$). As shown in Figure 3a, 3b, Lagrange gives us the exact result for $S\le7$. However, for larger values of $S$ and also in Figures 3c and 3d, the proposed approach, without any parameter tuning, outperforms Lagrange coded computing both in terms of computational stability (low variance) and the accuracy of recovery.

* The **computational complexity** of encoding and decoding in Lagrange Coded Computing are $\mathcal{O}(K\cdot \log^2(K) \cdot \log\log (K)\cdot d)$ and $\mathcal{O}((N-S)\cdot \log^2((N-S)) \cdot \log\log ((N-S))\cdot m)$, respectively, where $d$ and $m$ are input and output dimensions of the computing function $f(\cdot)$, respectively [2]. In contrast, for Smoothing Splines, the encoding and decoding process, which involves the evaluation of new points and calculating the fitted coefficients, have the computational complexity of $\mathcal{O}(K.d)$ and $\mathcal{O}((N-s).m)$, respectively, leveraging the B-Spline basis functions [5-7]. Consequently, the computational complexity of the proposed scheme is less than Lagrange Coded Computing.
Note that based on the BACC paper [1], the computational complexity of the Berrut method for encoding and decoding is the same as LeTCC. From the experimental view, we compare the whole encoding and decoding time (on a single CPU machine) for LeTCC and BACC frameworks, as shown in the following table:

||BACC|LeTCC
|-|-|-
|LeNet5, $(N,K)=(100, 20)$| $0.013s \pm 0.002$|$0.007s \pm 0.001$
|RepVGG, $(N,K)=(60, 20)$| $1.62s \pm 0.18$|$1.59s \pm 0.14$
|ViT, $(N,K)=(20, 8)$|$1.60s \pm 0.28$|$1.74s \pm 0.29$

We will include the above discussion in the revised version of the paper.

---

### Author Response · Authors · 2024-08-07
**References in the General Response to Reviewers**

[1]  Jahani-Nezhad, T., et al. "Berrut approximated coded computing: Straggler resistance beyond polynomial computing." IEEE PAMI 2022.

[2] Yu, Q., et al. "Lagrange coded computing: Optimal design for resiliency, security, and privacy." PMLR 2019.

[3] Yu, Q., et al. "Polynomial codes: an optimal design for high-dimensional coded matrix multiplication." NeurIPS 2017.

[4] Gautschi, W., et al. "Lower bounds for the condition number of Vandermonde matrices."" Numerische Mathematik 1987.

[5] Eilers, P.H., et al. "Flexible smoothing with B-splines and penalties." Statistical science 1996.

[6] De Boor, Carl. "Calculation of the smoothing spline with weighted roughness measure." Mathematical Models and Methods in Applied Sciences 11.01 (2001): 33-41.

[7] Hu, C.L., et al. "Complete spline smoothing." Numerische Mathematik 1986.

---

### Decision · Program_Chairs · 2024-09-25

**Decision:**

Accept (poster)

**Comment:**

This paper provides a new perspective on "coded computing" based on ML methods. The criticism seems to be directed at the viability of "coded computing" itself, rather than the paper. The implementation of the method is also questionable due to potentially excessive training time. The authors will do well to compare fast coded computing methods such as based on LDPC codes (Maity et al.), with their scheme.